# CD8+ T cells control SIV infection using both cytolytic effects and non-cytolytic suppression of virus production

Benjamin B. Policicchio[1,7], Erwing Fabian Cardozo-Ojeda [2,7], Cuiling Xu[3], Dongzhu Ma[3], Tianyu He[3], Kevin D. Raehtz[4], Ranjit Sivanandham [3], Adam J. Kleinman[4], Alan S. Perelson [5], Cristian Apetrei [1,4], Ivona Pandrea [1,3] & Ruy M. Ribeiro [5,6] ✉

Whether CD8+ T lymphocytes control human immunodeficiency virus infection by cytopathic or non-cytopathic mechanisms is not fully understood. Multiple studies highlighted non-cytopathic effects, but one hypothesis is that cytopathic effects of CD8+ T cells occur before viral production. Here, to examine the role of CD8+ T cells prior to virus production, we treated SIVmac251-infected macaques with an integrase inhibitor combined with a CD8-depleting antibody, or with either reagent alone. We analyzed the ensuing viral dynamics using a mathematical model that included infected cells pre- and post- viral DNA integration to compare different immune effector mechanisms. Macaques receiving the integrase inhibitor alone experienced greater viral load decays, reaching lower nadirs on treatment, than those treated also with the CD8-depleting antibody. Models including CD8+ cell-mediated reduction of viral production (non-cytolytic) were found to best explain the viral profiles across all macaques, in addition an effect in killing infected cells pre-integration (cytolytic) was supported in some of the best models. Our results suggest that CD8+ T cells have both a cytolytic effect on infected cells before viral integration, and a direct, non-cytolytic effect by suppressing viral production.

Understanding the host immune response against HIV/SIV infection is essential for developing effective therapeutic and preventive strategies. Unfortunately, HIV continuously evades and subdues the host's immune responses, muddling our attempts at elucidating the nature of the immune mechanisms needed to control infection. Examples of HIV evasion strategies include: (i) undergoing rapid mutation of its proteins due to host immune pressures to effectively evade host adaptive responses[1–3]; (ii) inducing down regulation of MHC-I expression through the viral protein Nef to reduce host cytotoxic capabilities to target infected cells[4]; (iii) taking advantage of virus-specific adaptive responses that generate activated CD4+ T cells, the preferential target of HIV, to propagate the infection[5]; (iv) chronically stimulating the immune system, thus resulting in production of nonfunctional exhausted cytotoxic T lymphocytes (CTLs)[6,7]. As such, there is

[1]Department of Infectious Diseases and Microbiology, Graduate School of Public Health, University of Pittsburgh, Pittsburgh, PA 15261, USA. [2]Vaccine and Infectious Disease Division, Fred Hutchinson Cancer Research Center, Seattle, WA 98109, USA. [3]Department of Pathology, School of Medicine, University of Pittsburgh, Pittsburgh, PA 15261, USA. [4]Division of Infectious Diseases, Department of Medicine, School of Medicine, University of Pittsburgh, Pittsburgh, PA 15261, USA. [5]Theoretical Biology and Biophysics Group, Los Alamos National Laboratory, Los Alamos, NM 87545, USA. [6]Laboratório de Biomatemática, Faculdade de Medicina da Universidade de Lisboa (previous address), Lisboa, Portugal. [7]These authors contributed equally: Benjamin B. Policicchio, Erwing Fabian Cardozo-Ojeda. ✉e-mail: ruy@lanl.gov

uncertainty regarding which immune response should be emphasized in current research efforts.

Of the various immune responses against HIV, the response exerted by CD8[+] T cells has been shown to be critical for control of HIV/SIV[8]. This is supported by: (i) the temporal association that exists between the increase in virus-specific CD8[+] T cell responses and the post-peak decline in plasma viremia[9,10]; (ii) the CD8[+] CTLs ability to suppress new infections in vitro[11,12]; (iii) the virus escape mutations that consistently arise in response to the host CD8[+] T cell response during all stages of infection[1,2,13,14]; (iv) the strong association between specific host MHC-I alleles and HIV/SIV disease progression[15]; and (v) the association of circulating escape mutants with the prevalence of specific HLAs in the population[16,17]. Experimental in vivo CD8[+] cell depletion studies in SIV-infected macaques have strengthened this argument and provided more direct evidence for the role of CD8[+] cells in HIV infection[18–28]. CD8[+] cell depletion results in a rapid and sustained rebound of plasma viremia, which is controlled when CD8[+] cells are restored. These results are consistent in many models of SIV infection: elite controller[18,29], nonpathogenic[30], rapid progressor[21,31], antiretroviral treated[20,22,23] and untreated models[19,24,26,32]. Accordingly, understanding the mechanisms of action of CD8[+] cells and identifying strategies to boost CD8[+]-specific immune responses is a key priority, both for HIV vaccine and cure research.

Although CD8[+] cells hold strong potential for cure efforts, their specific mechanism(s) of action is not well understood[8]. CD8[+] CTLs could exert a direct cytotoxic response against viral-infected cells via release of granzyme/perforin and/or stimulation of the Fas/FasL pathway[33,34]. Alternatively, CD8[+] cells could act by interfering with de novo infection or the release of new virions through soluble antiviral factors, including the CCR5-binding proteins MIP-1α, MIP-1β, RANTES, the cellular anti-HIV factor (CAF), α-defensins, and other factors[35–37]. To help shed light on this question, two groups studied the lifespan of SIV-infected cells after nucleotide reverse transcriptase inhibitor (NRTI) treatment, either in the presence or absence of CD8[+] cells. They showed that the average lifespan was not different with or without CD8[+] cells, concluding that CD8[+] CTLs do not exert a cytolytic effect on infected cells[20,23]. Another study quantified the lifespan of infected cells after antiretroviral treatment in infected people with different HLA background, both favorable and unfavorable for HIV progression[38]. These studies found no difference in the lifespan of infected cells and concluded that protective CD8[+] T cells may exert their effect before onset of productive infection, or via noncytolytic mechanisms, but none of them directly demonstrated this[20,23,38]. These data have both been corroborated and challenged[39–43], leaving the field to question what the true mechanism(s) of action of CD8[+] T cells is against HIV.

To help settle this question, we interrogated whether CD8[+] cells exert a cytolytic response against infected cells prior to viral production (i.e., before viral DNA integration into the host genome). The hypothesis is that once viral integration occurs and the cell starts producing virus, viral cytopathic effects dominate. Although this hypothesis has been proposed[20,42], it has never been tested experimentally. Here, to test this hypothesis, we administered the integrase inhibitor raltegravir (RAL) to SIV-infected rhesus macaques (RMs), in the presence or absence of CD8[+] cells. To analyze this data, we developed new viral dynamic models to account for CD8 depletion, adapted from our previous model[44], and fitted them to the data to study the possible effector mechanisms contributing to the observed viral load profiles. We found that the half-life of infected cells before viral integration in the RAL-treated only group is significantly shorter than in the RAL-treated plus CD8[+] depleted group, suggesting that CD8[+] T cells have a cytolytic role prior to viral integration. Further, the best models also indicated that the viral production rates increased in the absence of CD8[+] cells, indicating that CD8[+] T cells also exert a noncytolytic effect in suppressing viral production.

## Results

### Experimental design and mathematical modeling approach

Twenty (20) Indian-origin rhesus macaques (RM) were IV-infected with 300 TCID50 of SIVmac251 (Fig. 1). The initial dynamics of the virus were similar in all animals (Fig. S1). At fifty-six (56) days post-infection (dpi), after the virus reached quasi steady-state, CD8[+] cells were depleted by administration of the monoclonal antibody M-T807R1 to 12 RMs. Three weeks later, these animals received an additional M-T807R1 infusion. In 8 of these RMs, 2 days following the first CD8[+] cell depletion, RAL monotherapy was initiated for 23 days (CD8 depletion plus RAL Tx group, Fig. 1a). The remaining 4 RMs served as untreated controls (CD8 depletion group, Fig. 1a). An additional 8 RMs without CD8[+] T cell depletion were also treated with RAL monotherapy under the same conditions (RAL Tx group, Fig. 1a).

We sought to assess whether (i) RAL treatment combined with CD8[+] cell depletion would produce different dynamics for plasma viral load (pVL) and proliferating CD4[+] T cells in blood when compared to the RAL-only group; and whether (ii) these differences relate to a specific mechanism of CD8[+] cells in interfering with viral replication. To this end, we developed alternative mathematical models to help interpret the differences observed in the viral load profiles for each treatment group. We adapted our previously proposed slow and rapid virus integration (SRI) model[44], and fitted it to the viral load and proliferating Ki-67[+] CD4[+] T cell data from each RM group from the moment of the first M-T807R1 infusion until the end of RAL monotherapy (Fig. 1b). With this model, we explored a combination of possible CD8-depletion effects: (a) reducing the death rate of infected cells before SIV DNA integration into the host cell genome, (b) reducing the death rate of productively infected cells after SIV DNA integration, (c) increasing the rate of virus infection or (d) increasing virus production rate.

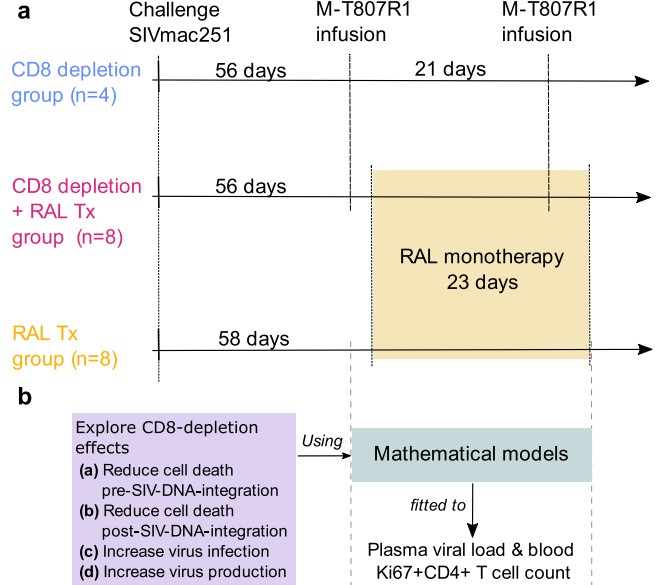

**Fig. 1 | Study approach. a** Experimental design: 20 rhesus macaques (RMs) were infected with SIVmac251. At 56 days post infection the CD8+ cells depleting antibody, M-T807R1, was infused in 12 RMs, with a second infusion three weeks later. Two days after the first CD8[+] cell depletion, raltegravir (RAL) monotherapy was initiated in 8/12 RMs (CD8 depletion group plus RAL Tx group), with the remaining 4/12 serving as untreated controls (CD8 depletion group). An additional 8 RMs without CD8[+] T cell depletion were also treated with RAL monotherapy under the same conditions (RAL Tx group). **b** We developed mathematical models to analyze plasma viral load and blood Ki67[+] CD4[+] T cell count from the time of first M-T807R1 administration to the end of RAL Tx.

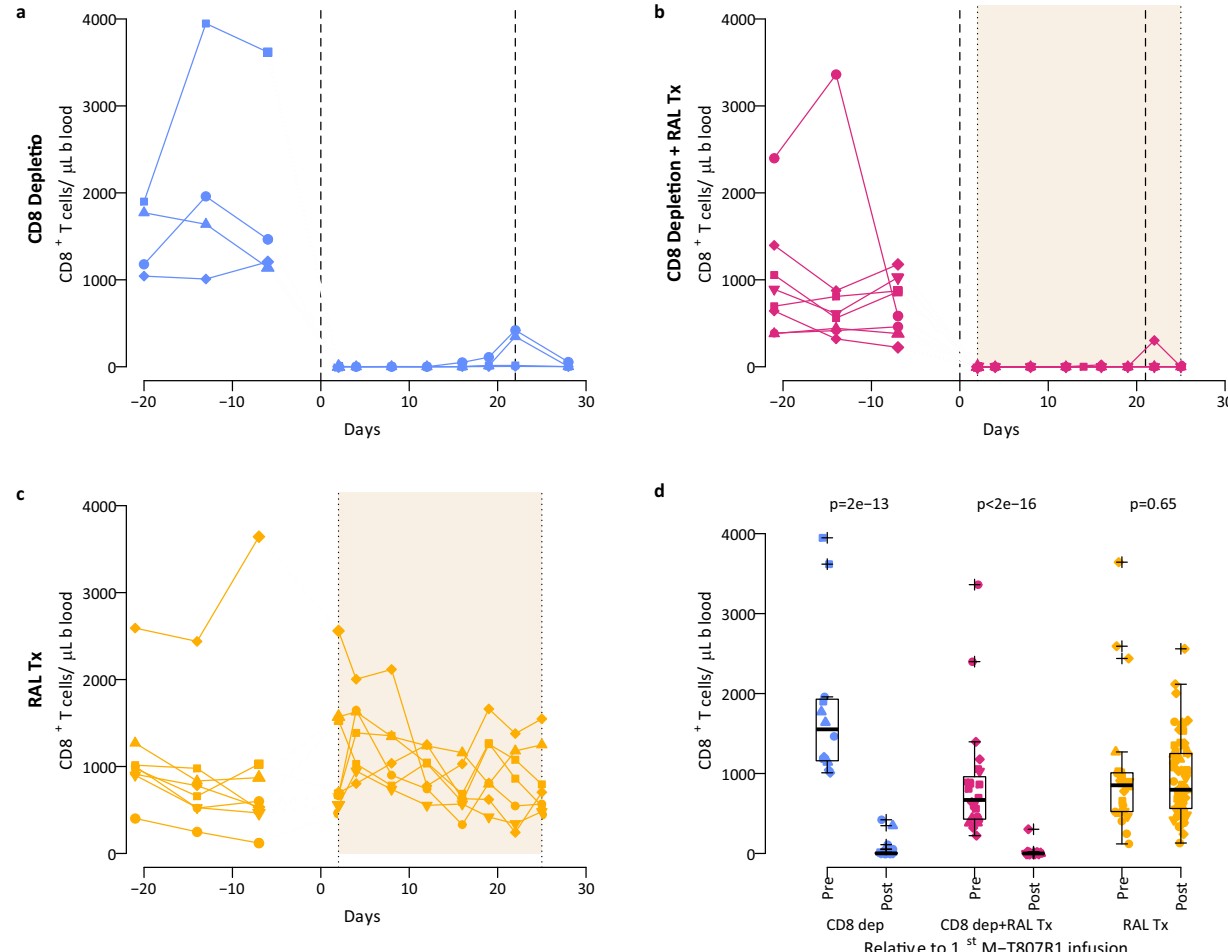

**Fig. 2 | The effect of M-T807R1 administration and RAL monotherapy on blood CD8+ T cell dynamics.** The effects of M-T807R1 on the dynamics of CD8 cells in individual macaques of **a** CD8 depletion group ($n = 4$), **b** CD8 depletion plus RAL Tx group ($n = 8$), and **c** RAL Tx group ($n = 8$). Vertical dashed lines represent the times of M-T807R1 administration and shaded regions represent the time of RAL monotherapy. **d** Distributions of the blood CD8+ T cell counts before (after viral load reached steady state) and after the time of first M-T807R1 administration. We used two-sided linear mixed effects to compare the repeated measurements pre- and post-infusion for the $n = 4$ (CD8 depletion group), $n = 8$ (each for CD8 depletion

plus RAL Tx group, and RAL Tx group). In all the panels, blue for CD8 depletion group; red for CD8 depletion plus RAL Tx group; and gold for RAL Tx group; with different symbols indicating different macaques. In (**d**) individual data points and box plots are presented. Box plots represent the 25th and 75th percentiles (bottom and top edge of the box), the median (line across the box), whiskers extending from the edge of the box to the smallest (bottom) or largest (top) value no further than 1.5 times the interquartile range from the box's edges, and extreme values beyond that shown as crosses.

## M-T807R1 effectively depletes CD8+ T cells from blood

In the 4 RMs in the CD8+ depletion group upon administration of the antibody, we saw a dramatic and sustained reduction in the numbers of circulating CD8+ T cells (average >99% reduction, $p < 0.0001$, Fig. 2a, d). CD8+ T cells remained depleted, with only a small transient recovery (average <197 cells/µL of blood) being observed at 21 days post depletion treatment, around the time of the second administration of M-T807R1 (Fig. 2a). We also observed a depletion of CD8+ T cells in tissues, both gut and lymph nodes (Fig. S2). Circulating NK cell levels decreased following M-T807R1 administration (average ~99% reduction) and remained suppressed throughout follow-up in most animals (Fig. S3).

The effect of the M-T807R1 infusion on the CD8 depletion group under RAL monotherapy was very similar to that described above for the control group, with rapid and effective reduction of the absolute count of circulating CD8+ T cells (average >99% reduction, $p < 0.0001$, Fig. 2b, d). This was also observed for the CD8+ T cells in the gut, but in lymph nodes the depletion was less pronounced in this group of macaques (Fig. S2). As expected, in the non-depleted group (RAL Tx group), CD8+ T cells did not decrease in blood or tissues, even showing

a small transient increase with the start of RAL in some macaques (Fig. 2c, d and Fig. S2). Overall, these observations are consistent with previous studies of CD8+ cell depletion in SIV infection[20–23,25,26,28].

## CD4+ T cells dynamics after M-T807R1 administration and RAL monotherapy

Upon the first round of M-T807R1 administration, peripheral CD4+ T cells significantly decreased in the CD8 depletion group (average 85% reduction in CD4+ T cells, $p < 0.0001$, Fig. 3a, d). Concomitantly with this reduction, a small and variable increase in CD4+ T cell proliferation occurred in some monkeys, as shown by the increase in the fraction of CD4+ T cells expressing the proliferation marker Ki-67. Then, CD4+ T cell proliferation gradually declined (Fig. 3a). In 3 of the 4 animals in the CD8 depletion group, the CD4+ T cell levels did not return to the levels prior to M-T807R1 infusion. The one macaque that reached pre-CD8-depletion levels of CD4+ T cells also had the lowest baseline level pre-depletion and showed larger fluctuations in Ki-67 expression.

The dynamics of CD4+ T cells were different between the CD8 depletion plus RAL Tx group and the RAL Tx only group. In the former group of RMs, there was an initial decline in CD4+ T cells after

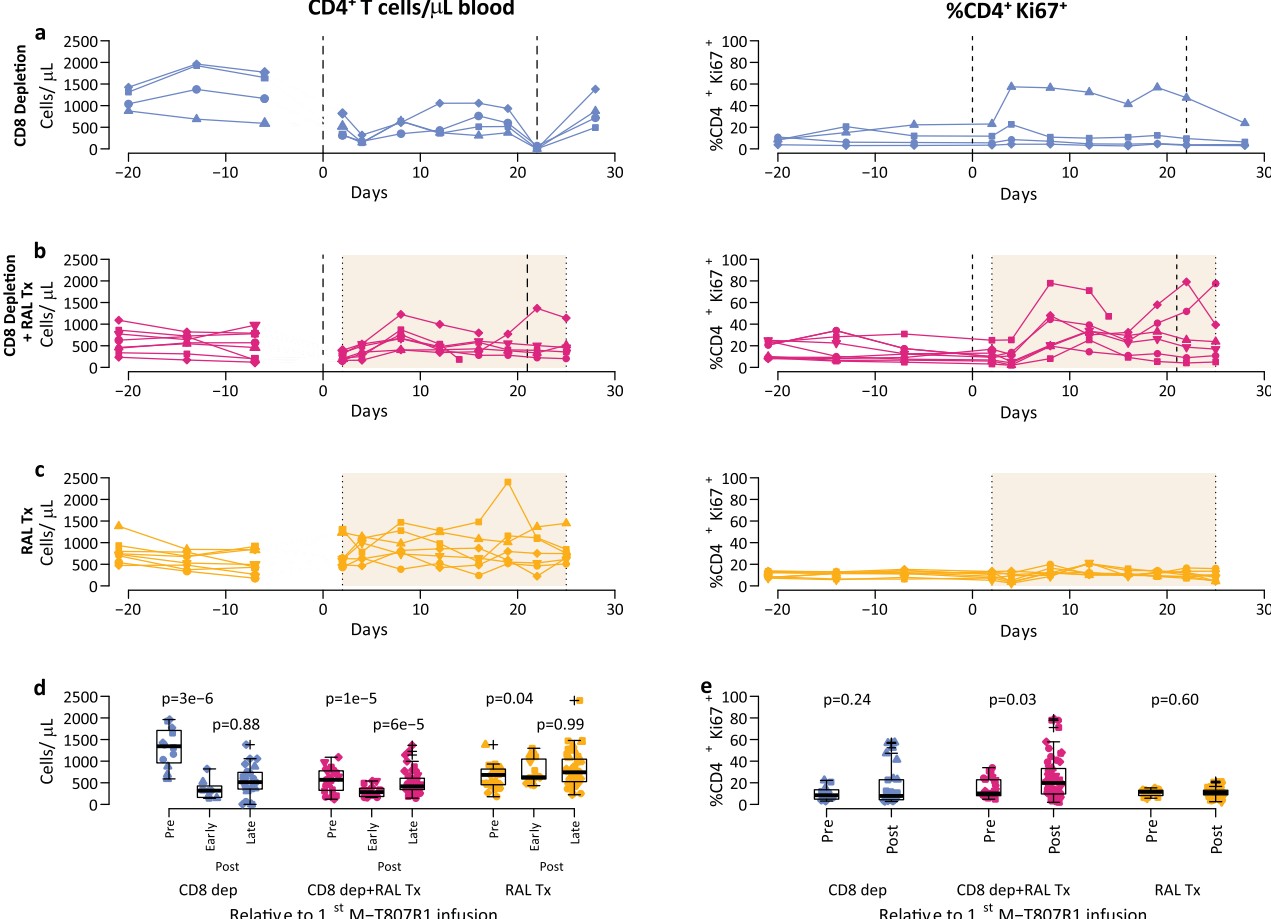

**Fig. 3 | The effect of M-T807R1 administration and RAL monotherapy on blood CD4⁺ T cell dynamics.** The dynamics of the peripheral blood CD4⁺ T cell count (left) and %Ki67⁺ CD4⁺ T cells (right) for individual macaques in (**a**) CD8 depletion group ($n = 4$), **b** CD8 depletion plus RAL Tx group ($n = 8$) and **c** RAL Tx group ($n = 8$). Vertical dashed lines represent the times of M-T807R1 administration and shaded regions represent the time of RAL monotherapy. **d** Distributions of the blood CD4⁺ T cell counts before the first M-T807R1 administration, at day 2–4–"Post early"– and after 4 days–"Post late"–relative to the first M-T807R1 administration. e. Distributions of the blood %Ki67⁺ CD4⁺ T cells before and after the first M-T807R1 administration. In d and e, we used two-sided linear mixed effects to compare the repeated measurements pre- and early post-infusion and to compare early post-

with late post-infusion (without adjustment for multiple comparisons in (**d**)), for the $n = 4$ (CD8 depletion group), $n = 8$ (each for CD8 depletion plus RAL Tx group, and RAL Tx group). In all the panels, blue for CD8 depletion group; red for CD8 depletion plus RAL Tx group; and gold for RAL Tx group; with different symbols indicating different macaques. In d and e, individual data points and box plots are presented. Box plots represent the 25th and 75th percentiles (bottom and top edge of the box), the median (line across the box), whiskers extending from the edge of the box to the smallest (bottom) or largest (top) value no further than 1.5 times the interquartile range from the box's edges, and extreme values beyond that shown as crosses.

depletion, as described above for the CD8 depletion only group, up to the start of RAL therapy (average 51% reduction in CD4⁺ T cells, $p < 0.0001$, Fig. 3b, d). This was followed by a recovery and stabilization throughout follow-up to levels similar to or even higher than prior to M-T807R1 administration (Fig. 3b, d). The proliferating fraction of CD4⁺ T cells did not immediately change in the CD8 depletion plus RAL Tx group; but during post-depletion the percentage of Ki-67⁺ CD4⁺ T cells did increase noticeably (median from 9 to 26%, $p = 0.03$, Fig. 3b, e), up to 17 days post-treatment when they stabilized with some variability (Fig. 3b). In the RAL Tx group, there was no decrease in peripheral CD4⁺ T cells, on the contrary, with treatment initiation, these cells increased, and then remained relatively stable, with minor variability throughout RAL treatment (Fig. 3c, d). Consistent with the total CD4⁺ T cell count in the RAL Tx group, CD4⁺ T cell proliferation experienced only a slight decline (average 2.4% decrease by 2 dpt, range: −2.1–9%) followed also by a small increase (average 7.2% by 10 dpt, range: −1.3–18.9%, Fig. 3c, e).

In conclusion, M-T807R1 treatment in SIVmac251-infected macaques leads to reductions in CD4⁺ T cells, which are accompanied by increases in CD4⁺ T cell proliferation, and eventual rebound in cell

numbers during RAL monotherapy. On the other hand, there was no important change in the RAL Tx group. We next analyzed the effect on viral dynamics of RAL therapy with and without CD8⁺ cell depletion.

## CD8⁺ T cell depletion induces an increase in plasma viral load and decreases the virological effect of RAL monotherapy

After the first M-T807R1 infusion, an increase of the plasma viral load (pVL) levels occurred in the CD8 depletion group, with a median increase of 0.89 $\log_{10}$ (range: 0.73–1.34 $\log_{10}$, Fig. 4a, d, e).

The pVL dynamics were different between the depleted and non-depleted groups under RAL monotherapy. First, we observed a median 0.9 $\log_{10}$ (range: 0.7–1.90 $\log_{10}$, Fig. 4b) increase in pVL post antibody administration in the CD8 depletion plus RAL Tx group. This viral load increase post CD8⁺ cell depletion was significantly different from the changes in pVL levels in the RAL Tx group ($p = 0.0001$, Fig. 4b, c, e). Upon initiation of RAL, pVLs decreased in the CD8⁺ depletion plus RAL Tx group, by a median of only 0.84 $\log_{10}$ (range: 0.22–1.78 $\log_{10}$, Fig. 4b, f). In stark contrast, in the RAL Tx group, pVL experienced a robust multiphasic decline following RAL initiation, with a median decrease of 1.75 $\log_{10}$ observed during treatment (range: 0.62–3.99

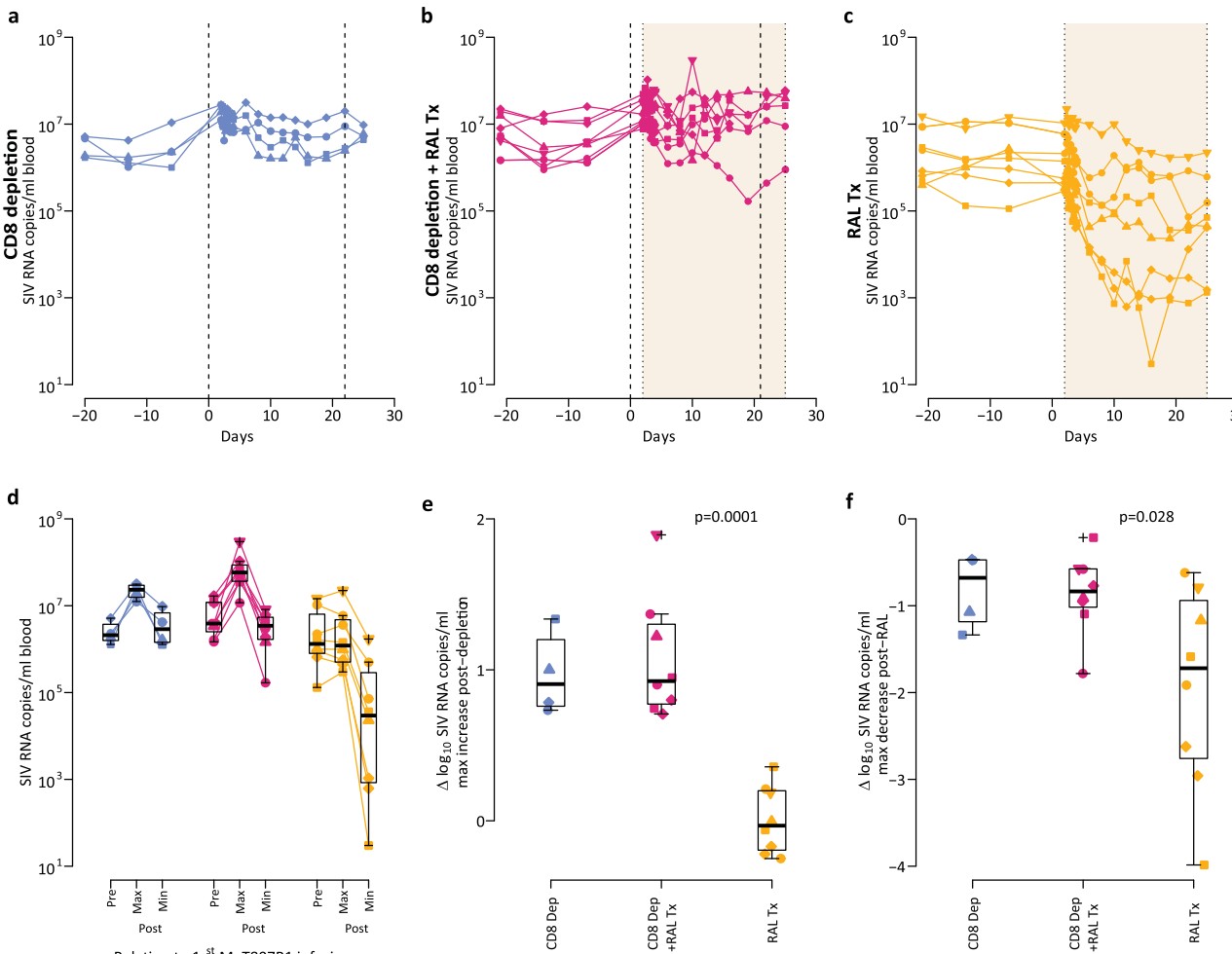

**Fig. 4 | The effect of M-T807R1 administration and RAL monotherapy on plasma viral load. a** CD8 depletion group ($n = 4$); **b** CD8 depletion plus RAL Tx group ($n = 8$); **c** RAL Tx group ($n = 8$). Vertical dashed lines represent the times of M-T807R1 administration and shaded regions represent RAL administration. **d** Distributions of the plasma median viral load before (after viral load reached steady state) the first M-T807R1 administration, and the maximum and minimum viral loads after (CD8 depletion group, $n = 4$; CD8 depletion plus RAL Tx group, $n = 8$; RAL Tx group, $n = 8$). **e** Distribution of the change in the $\log_{10}$ plasma viral load from the time before the first M-T807R1 administration (day 0 in panels **a, b, c**) to the maximum value after that time. **f** Distribution of the maximum decay in the $\log_{10}$ plasma viral load during the time of RAL monotherapy relative to the start of RAL. In (**e**) and (**f**), we used the Mann–Whitney test (two-sided) to compare the CD8 depletion plus RAL Tx group with the RAL Tx group ($n = 8$ for each group). In all the panels, blue for CD8 depletion group; red for CD8 depletion plus RAL Tx group; and gold for RAL Tx group; with different symbols indicating different macaques. In (**d**) to (**f**), individual data points and box plots are presented. Box plots represent the 25th and 75th percentiles (bottom and top edge of the box), the median (line across the box), whiskers extending from the edge of the box to the smallest (bottom) or largest (top) value no further than 1.5 times the interquartile range from the box's edges, and extreme values beyond that shown as crosses.

$\log_{10}$, Fig. 4c, f), a significantly larger decrease than for animals in the CD8 depletion plus RAL Tx group ($p = 0.028$, Fig. 4f).

In conclusion, CD8$^+$ T cell depletion induces an increase in pVL and decreases the virological effect of RAL monotherapy. We next tried to understand the mechanistic basis for these clear differences in plasma viral dynamics upon RAL treatment in the depleted vs. non-depleted groups. To that end, we explored different hypothesis for the effect of CD8$^+$ cells using mathematical models of viral dynamics.

## CD8$^+$ cells kill infected cells before SIV DNA integration and reduce viral production

We used the model in Eq. (1) (see "Methods" and Fig. 5a for a description of the model) to help interpret the differences observed in the virus dynamics for each treatment group. We analyzed different combinations of four possible assumptions previously suggested for the effects of CD8$^+$ cells[11,20,23,26,38,42,45,46], i.e., CD8$^+$ cells could have cytolytic effects and (i) kill infected cells before integration and/or (ii) kill productively infected cells (i.e., after integration). CD8$^+$ cells could

have non-cytolytic effects (e.g., transcriptional/ translational silencing) or secrete factors (e.g., chemokines and cytokines) that (iii) reduce the rate of viral infection, or (iv) reduce the rate of viral production. Depletion of CD8$^+$ cells potentially affects each of these mechanisms in the model. In addition, given our observations above (and previous work[20]) that CD4$^+$ T cell proliferation increases with the depletion treatment, we also included this effect of depletion in our model.

We then fitted 32 instances of the model (with or without each of those five effects of depletion, thus $2^5$) in Eq. (1) to the viral load and Ki67$^+$ CD4$^+$ T cell data from all the RMs in the three treatment groups simultaneously, using a mixed-effects approach (see Methods). In each instance of the model fit, we allowed a different combination of the five mechanisms described above (Fig. 5b) to see what effects of CD8$^+$ cell depletion best explained all the data. Using model selection theory, we found strong support for two effects of CD8$^+$ cell depletion: (i) it increases CD4$^+$ T cell proliferation and (ii) it increases the rate of virus production ($p$). All the best models consistently required these two effects. In addition, we also found support for (iii) a decrease in the

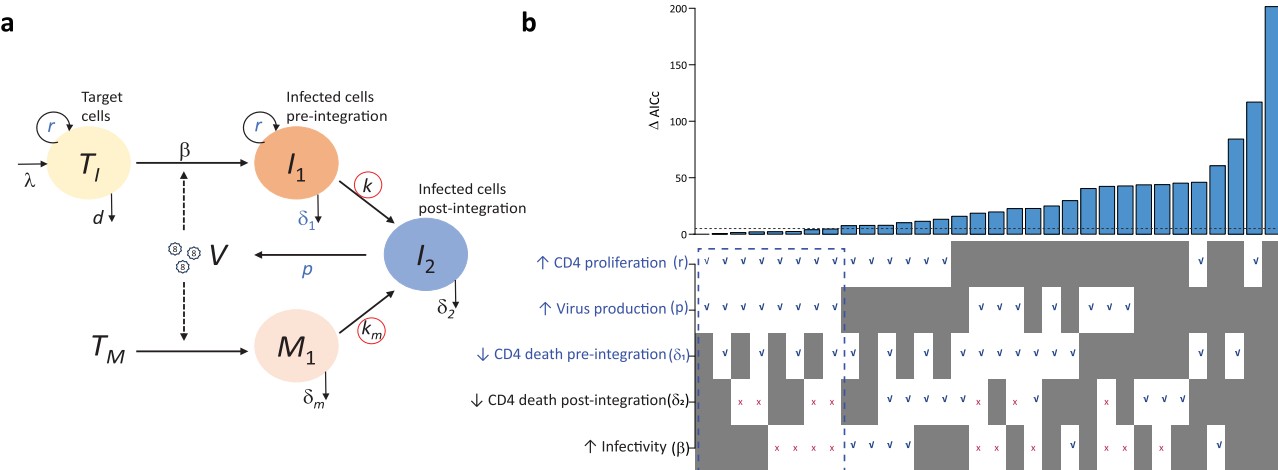

**Fig. 5 | Mathematical model schematics and CD8+ cell effects modeled.**
**a** Diagram describing the viral dynamics mathematical model. $T_I$ represents pro-liferating target cells that after infection will be short-lived pre-integration infected cells $I_1$. $T_M$ represents a constant level of target cells that after infection become long-lived pre-integration infected cells $M_1$. $I_2$ represents infected cells after SIV DNA integration that produce virus $V$. Parameters inside red circles, represent the parameters affected by raltegravir. Effects/parameters in blue are those selected by the model selection procedure. Other details are described in "Methods".
**b** Difference in the corrected Akaike Information Criteria (ΔAICc) of each model instance in relation to the one with lowest AICc. Models with ΔAICc<5 represent the most parsimonious models (dashed blue box below the plot). The 32 model instances (represented by the bars in the plot and the columns in the table below the plot) codify a combination of effects that may occur under CD8+ cell depletion: (i) increase in CD4+ T cell proliferation, (ii) reduction in the viral production rate, (iii) reduction in the infected cell death rate pre-integration, (iv) reduction in the pro-ductively infected cell death rate, or (v) increase in the viral infection rate. Gray patches in the table represent effects that *were not* tested in that model instance (in each column). Blue check marks indicate that the corresponding effect was sig-nificant in that model and red cross represent effects that were tested but not significant in that model instance.

death rate of infected cells prior to SIV DNA integration ($\delta_1$ in the model), which was observed in several of the top best models (Fig. 5b, Table S1). In contrast, there was no support for an effect of CD8+ cell depletion on the death of infected cells post-integration ($\delta_2$) or on virus infectivity (β).

Figure 6 and Figure S4 show the best fits of the model to the viral load and CD4+ Ki67+ T cell data, respectively, using the individual parameter estimates in Table S2. From the best fits, our model predicts that CD8+ cell depletion leads to (i) a CD4+ T cell proliferation rate that is up to five times faster; and (ii) to a virus production rate increase of 1.8-fold (Table 1). The top best models also included a reduction in the pre-integration infected cell death rate (Table S1). Without CD8+ cell depletion, our model predicts that productively infected cells have a median half-life of ~12 h ($\delta_2$ ~ 1.4 day$^{-1}$) and short-lived infected cells before SIV DNA integration have a median half-life of ~4.9 days ($\delta_1$ ~ 0.14 day$^{-1}$) characterizing what have previously been called phases 1a and 1b of the viral decline during RAL monotherapy[44] (Table 1). Finally, our model also predicts that the median efficacy of RAL is 94%, except for animals RM238 and RM239 where this efficacy is 66% (Table 1). Although we do not know why RAL showed less efficacy in these macaques, their viral loads decay less upon treatment than in the other macaques.

To further test our results and compare to previous work[20,23,42], where CD4+ T cell proliferation was not accounted for, we also fitted the data with a simpler model that did not account for CD4+ T cell proliferation. We obtained similar results in terms of the effects of CD8+ cell depletion, namely an increase in viral production of ~2-fold and a decrease in the death of infected cells pre-integration of ~75%, without any noticeable effect on death of cells productively infected or infection rate (see Supplementary Note 3, Figs. S5, S6 and Table S3).

In Figs. S7 and S8, we show the profile likelihoods for the fitted parameters of both models (see Supplementary Note 3). These show that most of the parameters are identifiable in both models, except for the effect of CD8+ depletion on death of infected cells pre-integration in the full model, for which the profile likelihood is flat.

## Discussion

CD8+ T cells are strongly associated with the control of HIV/SIV in various models of SIV infection and multiple progression scenarios of HIV and SIV infection[10,11,18,19,21,22,24-26,29,30]. They have been shown to contribute to the suppression of pVL in SIV-infected RMs on long-term ART[22]. Their control is exerted through either a direct, cytotoxic killing of the infected cell and/or by noncytotoxic mechanisms, whereby the release of chemokines and cytokines inhibits viral infection or enacts transcriptional silencing[35-37,46-48]. At least three papers, using two completely different approaches, independently concluded that CD8+ cells do not exert a cytolytic effect on productively infected cells and thus inferred that CD8+ cells mainly act via a noncytolytic response against HIV/SIV-infected cells[20,23,38]. However, the suggestion that CD8+ cells do not exert a cytolytic effect is not universally accepted[39,42,45,47], one possibility is that the CD8+ cells cytolytic effect after viral pro-duction starts is obscured by the larger cytopathic effect.

Our goal was to clarify the possible mode of action of CD8+ cells, namely their effect on SIV-infected cells after reverse transcription but before integration. With our new experimental system, by utilizing RAL, we can analyze specifically the effects of CD8+ cells on infected cells prior to viral integration, a phase of the SIV infection cycle not studied previously. We discovered that the decline of viral load during RAL monotherapy was indeed different in the presence and absence of CD8+ cells, which had not been seen before in other models testing the effect of CD8+ cells on treatment[20,23,38].

Having found this major difference, we next fitted novel mechanistic models of viral dynamics to the data and explored mul-tiple scenarios for the mechanisms of CD8+ cells' action to control virus: (i) killing infected cells prior to viral integration; (ii) killing infected cells following viral integration (i.e., productively infected cells); and noncytolytically reducing viremia through indirect mechanisms, such as (iii) reducing the rate of viral infection or (iv) reducing the rate of viral production. We used a mixed-effects fitting approach, which allowed us to use all the data simultaneously to achieve more robust results.

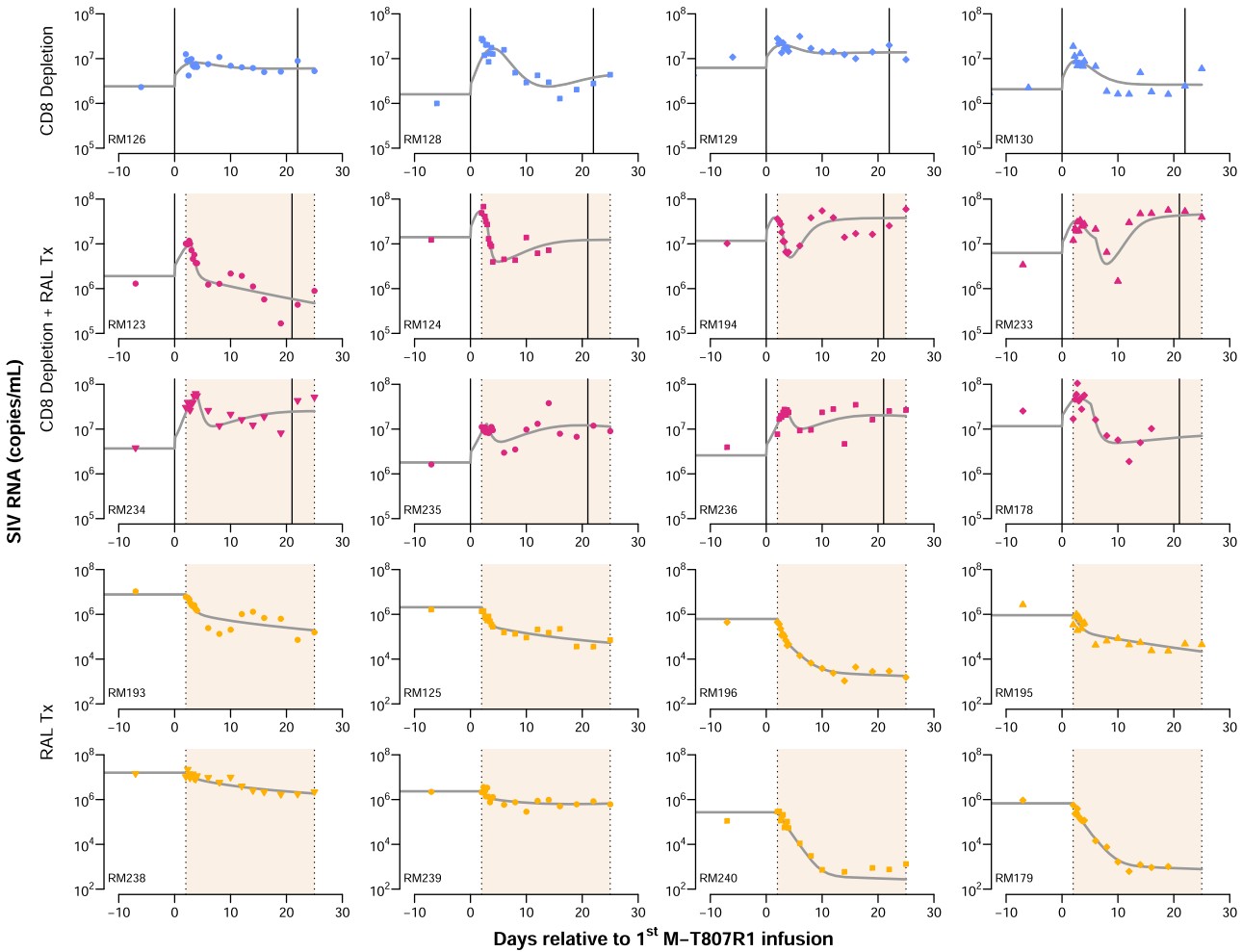

**Fig. 6 | Fits of the best model to the viral load data.** Each panel presents the viral load (symbols) dynamics of an individual macaque for the three study groups: CD8 depletion group (top row), CD8 depletion plus RAL Tx group (2nd and 3rd row), and RAL Tx group (bottom two rows). Vertical-solid lines represent the times of M-T807R1 administration and shaded regions represent the time of RAL monotherapy. Gray solid lines show the model fits using individual parameter estimates in Table S2.

The different decay characteristics between the two RAL-treated groups (CD8 depletion plus RAL Tx or RAL Tx) reflect the inherent ability of infected cells to present peptides derived from the infecting virus to cognate CD8⁺ cells. It has been shown that infected cells are able to present Gag-derived epitopes as early as 2 h post-infection[49] and that the first wave of antigen presentation occurs between 3 and 6 h post-infection of a cell[50], corresponding to the initial period of reverse transcription[51]. In a seminal experimental paper, Sacha et al.[49] used cells from Indian origin rhesus macaques (as in our study) in vitro to show that CD8+ T cells recognize Gag-derived epitopes from SIV infection very early after infection, with recognition peaking at 6 h post-infection, i.e., before viral integration. Moreover, they went further and showed that these epitopes are derived from the incoming virus, and directly showed that specific CD8 + T cells eliminate infected CD4+ T cells before 6 h post-infection[49]. In another paper, an independent group showed essentially the same results of early recognition, activation and killing by CD8+ specific T cells recognizing early presented epitopes in HIV infected human CD4+ T cells (and other targets) before viral integration and before protein synthesis[50]. These two independent and complex experimental studies demonstrate that CD8+ T cells can kill infected cells before SIV/HIV integration, and thus before viral production. The use of RAL in our study allows us to analyze in vivo the effects of direct CD8⁺ cell killing before viral integration, during the time when antigen presentation of viral proteins

such as Gag from the infecting virus is optimal while minimizing virus-induced cell death.

Our results support a dual effect of CD8⁺ cells, which seem to exert both a cytolytic effect against infected cells prior to viral integration, and a noncytolytic suppression of viral production by infected cells after integration. This is in partial contrast to previous results indicating a non-cytolytic effect[20,23,38], because we show that, in addition to such effect, CD8⁺ cells also act cytolytically during a small window of infection, before pro-viral integration. Those previous results by Klatt et al.[20] and Wong et al.[23] showed no measurable effect of CD8⁺ cell depletion on the pVL decay rate after ART initiation, because the NRTI therapy (and protease inhibitor therapy) used only quantifies the decay of productively infected cells[28]. However, following integration, as the cell starts producing virus, viral-induced cytopathic processes occur, obscuring and preventing the separation of the effect of CD8⁺ cells from viral cytopathic effects on infected cells. On the other hand, our results are compatible with a more limited analysis based on the dynamics of infected cells with 2-long-terminal repeats (2LTR) circles under RAL therapy[52]. We emphasize that there is not a discrete switch in the CD8⁺ cells' mode of action, from killing to inhibiting production. Rather the observed dual effect is due to the lifecycle of HIV: before integration there is no viral production so no inhibition by CD8⁺ cells can be observed, and cytolytic effects dominate; after integration it is possible that killing by viral pathogenic

**Table 1 | Estimated population parameters for the model with the best fit**

| Parameter | | Description | Fixed effects | | STD Random effects | |
|---|---|---|---|---|---|---|
| | | | $\theta$ | %RSE | $\sigma_\theta$ | %RSE |
| $f_i$ | | Fraction of virus produced by short-lived infected cells pre-CD8-depletion | 0.97 | 1.4 | 0.5 | – |
| $\log_{10} V_0$ [$\log_{10}$ copies/mL] | | Log10 Viral load pre-CD8-depletion | 6.5 | 1.8 | 0.5 | 16.5 |
| $r$ [1/day] | | Maximum proliferation rate of Ki67+CD4+ T cells | 0.02 | 39.5 | 0.9 | 36.8 |
| $K$ [cells/µL] | | Carrying capacity for proliferation rate of Ki67+CD4+ T cells | 76.8 | 79.8 | 2.9 | 20.6 |
| $\delta_1$ [1/day] | | Death rate of SIV-infected short-lived cells pre-SIV-DNA integration | 0.14 | 32.5 | 0.5 | – |
| $\delta_2$ [1/day] | | Death rate of productively SIV-infected cells | 1.4 | 14.5 | 0.5 | – |
| $\delta_m$ [1/day] | | Death rate of SIV-infected long-lived cells pre-SIV-DNA integration | 0.02 | – | 0.2 | – |
| $\omega$ | for majority | Efficacy of raltegravir in reducing SIV-DNA integration events | 94% | 0.9 | 0.2 | 92.2 |
| | for RM238, RM239 | | 66% ($p = 0.001$)* | 31 | | |
| $\log_{10}\beta$ [µL/day/mL] | | Log10 SIV infectivity rate | –7.8 | 1.1 | 0.3 | 25.3 |
| $p$ [1/day] | | Virus production rate | 40,000 | – | 0.5 | – |
| $\tau$ [days] | | Pharmacological delay of raltegravir | 1.1 | 34 | 0.9 | 28.6 |
| CD8 depletion effects | $\xi_1$ | Fraction reduction of death rate of SIV-infected short-lived cells pre-SIV-DNA integration during CD8-cell-depletion | 0 | – | – | – |
| | $\xi_2$ | Fraction reduction of death rate of productively SIV-infected cells during CD8-cell-depletion | 0 | – | – | – |
| | $1+\xi_3$ | Fold increase in virus infectivity during CD8-cell-depletion | 1 | – | – | – |
| | $1+\xi_4$ | Fold increase in virus production rate during CD8-cell-depletion | 1.84 | 11.7 | – | – |
| | $1+\xi_5$ | Fold increase in maximum proliferation rate of Ki67+CD4+ T cells | 4.9 | 34 | – | – |
| Measurement error | $\sigma_v$ | Measurement error for the logged viral load | 0.23 | 3.9 | – | – |
| | $\sigma_t$ | Measurement error for the logged Ki67+CD4+ T cell count | 0.26 | 5.3 | – | – |

*P-value computed by Monolix using Wald test (two-sided).

Note: %RSE: %Relative standard error calculated by Monolix. Parameters without an entry for %RSE were fixed and parameters without an entry for $\sigma_\theta$ did not have a random effect.

effects is too fast to observe the cytolytic effect of CD8[+] cells, and the inhibition of production effect is dominant.

The two rounds of M-T807R1 maintained suppression of CD8[+] cells throughout the treatment. The small, transient rebounds in CD8[+] cells observed during treatment did not affect the subsequent analysis, as these cells are most likely nonfunctional[19]. The observed increase in pVL following CD8[+] cell depletion in the animals that received M-T807R1 supports the role that CD8[+] cells play in controlling viremia. In this context, fitting the data of the CD8[+] cell depleted group without RAL therapy helped us to identify the effect of these cells on viral production $p$. We note that one of the first papers on CD8[+] cell depletion had mentioned this possibility, but they did not have enough data (e.g., the frequent sampling) for definitive conclusions[26]. Recent in vitro studies also suggested a non-cytolytic effect of CD8[+] T cells in preventing viral production, rather than reducing viral infection, which is consistent with our findings[46], with a second study investigating HIV latency finding that CD8[+] T cells induced changes in metabolic and signaling in CD4[+] T cells, leading to reduction in HIV expression[53]. Another possibility is that very early on in infection, such as during acute infection, CD8 cells are more functionally cytolytic, but this phenotype could change over time[8]. We note that we conducted the experiment as early as possible after acute infection, at the beginning of chronic infection when the virus reached quasi-steady state. Some authors have argued that in models of natural control of SIV infection, CD8[+] cytolytic mechanisms are preponderant[21,54]. Although those authors have not analyzed their data in as much detail as we do here, we can't exclude the possibility that our results apply mostly to non-controlled SIV infection, which is the relevant model to the majority of HIV cases.

The depleting antibody M-T807R1 binds to the CD8α receptor on cells and efficiently depletes CD8[+] T cells, as seen in Fig. 2a, b. It also depletes NK cells and NKT cells, both of which express CD8α on their surface. This represents a potential limitation of our study, as both NK and NKT cells exert antiviral effects during HIV infection[55,56]. Indeed, we see a depletion of NK cells in the blood following CD8[+] cell depletion (Fig. S3). Thus, we cannot be certain that the effects we see in viral load and that our model estimates are not due to NK or NKT cells in addition to CD8[+] T cells. However, it has been shown that NK cells upregulate inhibitory receptors during viremic HIV infection, affecting the killing ability of NK cells[57,58]. If that were the case, any effect of NK cells on viral replication would be minimal, regardless of CD8 depletion. Another potential limitation is the more restricted depletion of CD8[+] cells in lymph nodes in our RAL-treated animals, which is statistically significant (Fig. S2). However, if anything, this observation has an impact against our results, because we reached them in the absence of full depletion, indicating that our conclusions on the effect of CD8[+] cells are even stronger. Another potential factor that could bias our analysis is an increase in CD4[+] T cell activation following CD8[+] cell depletion[20], which would increase the availability of target cells leading to more infections and more viral production[27]. We accounted for this in our analyses by explicitly incorporating the proliferation of target cells in our models and fitting them to data on CD4[+] T cell proliferation, which was typically not done in previous studies[20,39,42]. Indeed, our results showed that an increase in CD4[+] T cell proliferation rate after CD8[+] cell depletion was a robust finding, in spite of the variability seen in the dynamics of Ki67 CD4+ T cells (Fig. S4). As a sensitivity analyses to our model formulation and to the use of this variable data, we also fitted a model only to the viral load without considering the Ki67 CD4+ T cell data (Fig. S5), and reached the same conclusions on the effect of CD8+ cells, namely that they exert a cytolytic effect before viral integration and non-cytolytic suppression of viral production.

In conclusion, our study definitively shows a strong non-cytolytic effect of CD8[+] cells suppressing viral production. In addition, it demonstrates that, prior to viral integration, CD8[+] cells do exert a direct cytolytic effect against infected cells. Our results expand on previously published data and greatly contribute to our understanding of the effects exerted by CD8[+] cells during HIV/SIV infection, by pinpointing the mechanisms of action. Our data will help inform future studies focused on both developing a vaccine to prevent new HIV infections and new cure strategies for those already infected.

## Methods

### Ethics statement
Rhesus macaques (RMs) were housed and maintained at the University of Pittsburgh, Plum Borough animal facility, according to the standard of the Association for Assessment and Accreditation of Laboratory Animal Care (AAALAC) International, and experiments were approved by the University of Pittsburgh Institutional Animal Care and Use Committee (IACUC), protocol #16058287. The animals were cared for according to the *Guide for the Care and Use of Laboratory Animals* and the Animal Welfare Act[59]. Efforts were made to minimize animal suffering: all RMs had 12/12 light/dark cycle, were fed twice daily with commercial primate diet, water was provided ad libitum, and were socially housed in pairs indoors in suspended stainless-steel cages. Various environmental enrichment strategies were employed: providing toys to manipulate and playing entertainment videos in the animal rooms. The animals were observed twice daily for any signs of disease or discomfort, any of which were reported to the veterinary staff for evaluation. At the end of the study, the animals were euthanized in accordance with the recommendations of the American Veterinary Medical Association (AVMA) Guidelines for the Euthanasia of Animals.

### Animals, infections, and treatments
Twenty (20) Indian-origin RMs (*Macaca mulatta*), 5–7 years old males, were included. These macaques had not been used before for any other studies. All macaques were IV-infected with 300 TCID50 of SIV-mac251 and were closely monitored during all stages of the study. Fifty-six days post-infection (dpi), after the virus reached quasi steady-state, the CD8[+] cell-depleting monoclonal antibody M-T807R1 (Cat# PR-0817, RRID:AB2716320, NIH Nonhuman Primate Reagent Resource, Boston, MA) was administered to 12 RMs at a dose of 50 mg/kg. The animals received an additional 10 mg/kg of M-T807R1 three weeks later. Two days following the first CD8[+] cell depletion, RAL monotherapy was initiated, at 20 mg/kg *bid* for 23 days, in 8 of these RMs, with the remaining four serving as untreated controls. An additional 8 RMs without CD8[+] cell depletion were also treated with RAL monotherapy under the same conditions (Fig. 1). Two RMs (RM124 and RM178), in the CD8[+] cell depletion with RAL treatment group, had to be euthanized at 70 and 72 dpi, respectively, due to AIDS-like manifestations. We note that the control group with depletion only was smaller ($n = 4$), because this simple depletion experiment has been done many times in the context of SIV infection, and for animal welfare issues (and cost) we just needed to make sure that our depletion was equivalent to historical controls, demonstrating that the protocol was working properly. Eight animals each for the two RAL-treatment groups, based on previous data on SIV-infected ART-treated rhesus macaques, allows detection of a difference in the level of plasma SIV RNA with an effect size of 2, at α=0.05 significance level with a power of at least 0.9.

### Sampling and sample processing
Blood was collected from all RMs twice prior to infection and then approximately weekly after infection up to treatment initiation. When viral loads reached a quasi-steady sate, defined by three consecutive stable VLs, treatment was started. After treatment initiation of RAL, blood was sampled every 6 h for 2 days, then every 2 days for 2 weeks and then every 3 days until 23 dpt. No data was excluded from the analyses.

Within one hour of blood collection, plasma was harvested and peripheral blood mononuclear cells (PBMCs) were separated from the blood using lymphocyte separation media (LSM, MPBio, Solon, OH).

## Plasma viral load (pVL) quantification

We monitored the levels of free, circulating virus at all times indicated above. Plasma from all animals was subject to a quantitative reverse-transcription PCR (qRT-PCR). The primer and probe sequences amplify a conserved region of Gag and are as follows: SIVmac251F: 5′-GTC TGC GTC ATC TGG TGC ATT C-3′; SIVmac251R: 5′-CAC TAG GTG TCT CTG CAC TAT CTG TTT TG-3′; SIVmac251Probe: 5′-CTT CCT CAG/ZEN/TGT GTT TCA CTT TCT CTT CTG CG/3IABkFQ/-3′. Real-time PCR was performed utilizing an ABI 7900 HT real-time machine (Applied Biosystems, Foster City, CA, and SDSv2.4.1 software) with the following parameters: 95 °C for 10 min, 45 cycles of 95 °C for 15 s, 60 °C for 1 min.

## Flow cytometry

Whole blood was stained at specific pre- and during treatment time points to monitor the impact of treatment on major immune cell populations. The two-step TruCount (BD Bioscience) technique was used to enumerate the absolute number of CD4+ and CD8+ T cells in blood[18,60]. Blood was stained with fluorescently-labeled antibodies (antibodies from BD Bioscience, San Jose, CA, USA, except where noted): CD4 (APC, cat# 551980, 2.5 µl), CD8 (PE-CF594, cat#: 562282, 3 µl), CD3 (V450, cat#: 560351, 2 µl), CD45 (PerCP, cat#: 558411, 3 µl), NKG2A (PE, Beckman Coulter Life Sciences, Indianapolis, IN, USA, cat#: IM3291U, 5 µl). In addition, Ki-67 (PE, cat#: 556027, 20 µl) was stained post fixing and permeabilizing cells after the surface staining. Flow cytometry acquisitions were performed on an LSR II flow cytometer (BD Biosciences, with FACSDiva v8.01, and FlowJo v10.4, v10.7.1). All antibodies were used following the manufacturers' recommendations, and validation for use of these antibodies in rhesus macaques was ascertained by the NIH (as indicated in www.nhpreagents.org/ReactivityDatabase).

## Mathematical modeling

To help interpret the viral load profiles during CD8+ cell depletion and RAL monotherapy, we adapted the slow and rapid integration (SRI) viral dynamic model proposed by Cardozo et al.[44], which is a modification of the standard model of virus dynamics[61] shown to be more appropriate to study integrase inhibitor treatment. In this model, Eq. (1), we follow two types of target cells, those that after infection will be short-lived ($T_I$) and those that will be long lived ($T_M$). The former, $T_I$, are created at a constant rate $\lambda$, die at rate $d$, and are infected by the virus ($V$) at rate $\beta$. These infected cells ($I_1$) are lost at rate $\delta_1$ possibly influenced by effector CD8+ cells and undergo integration of the proviral DNA at rate $k$, to become productively infected cells ($I_2$). We assume that target cells $T_M$ remain approximately constant during the ~3 weeks of the experiment and are infected by virus at rate $\beta_m$. This infection event produces long-lived infected cells ($M_1$) that are lost at rate $\delta_m$ or undergo provirus integration at a slower rate $k_m$ also becoming productively infected cells ($I_2$). An integrase inhibitor, such as RAL, blocks proviral DNA integration with efficacy $\omega$ after a pharmacological delay $\tau$. Cells with integrated HIV DNA ($I_2$) are productively infected and are lost at rate $\delta_2$. Virions are produced by these cells at rate $p$ per cell and are cleared from circulation at rate $c$ per virion. To account for the possible effect of the depletion antibody in inducing proliferation of CD4+ T cells we allow for proliferation of short-lived CD4+ T cells, so $T_I$ and $I_1$ expand logistically with maximum rate $r$ and carrying capacity $K$. We also assume that once cells become productively infected ($I_2$) proliferation is negligible, because death occurs

rapidly. The model equations are:

$$\frac{dT_I}{dt} = \lambda - dT_I - \beta T_I V + rT_I\left(1 - \frac{T_I + I_1}{K}\right)$$

$$\frac{dI_1}{dt} = \beta T_I V - k(1-\omega)I_1 - \delta_1 I_1 + rI_1\left(1 - \frac{T_I + I_1}{K}\right)$$

$$\frac{dM_1}{dt} = \beta_m T_M V - k_m(1-\omega)M_1 - \delta_m M_1 \qquad (1)$$

$$\frac{dI_2}{dt} = k_m(1-\omega)M_1 + k(1-\omega)I_1 - \delta_2 I_2$$

$$\frac{dV}{dt} = pI_2 - cV$$

In Supplementary Note 1, we present an alternative, simpler model that does not consider cellular proliferation. We note that in these models, we separate the viral lifecycle in pre- and post-integration stages, because integrase inhibitor therapy allows us to probe these stages, however they exist independently of treatment or the type of treatment. Indeed, before we used the same model to analyze data from people treated only with reverse transcriptase inhibitors[44].

## Data fitting and model selection

To fit the model to the data, we used nonlinear mixed-effect modeling as previously described[44]. We simultaneously fit the model to plasma viral load and Ki67+ CD4+ T cell observations from rhesus macaques in the three groups. We modeled the plasma viral load and the Ki67+CD4+ T cell count on a log10 scale for animal $i$ at time $j$ as $y_{ij} = \log_{10} V(t_j) + \epsilon_V$ and $z_{ij} = \log_{10}[T_I(t_j) + I_1(t_j)] + \epsilon_T$, respectively, with $\epsilon_V \sim \mathcal{N}(0, \sigma_v^2)$ and $\epsilon_T \sim \mathcal{N}(0, \sigma_t^2)$, the error for the logged viral load and Ki67+CD4+ T cell count. In this approach, we assume that a model parameter $\eta_i$ is drawn from a distribution (that may differ by treatment group) in the animal population with a fixed part $\theta$, which is the median value of the parameter in the population, and a random term $\phi_i$, which is assumed to be normally distributed with zero mean and standard deviation $\sigma_\theta$. Unless otherwise specified, we assumed that parameters follow a lognormal distribution. We fit instances of the model to the data and estimate the median and variances of the distribution of each parameter using the software Monolix versions 2020 R, 2021R1 (Lixoft SAS, Antony, France)[62].

Each instance of the model assumes that CD8+ cell depletion has one or more of the following effects: (i) reduction of the death rate of short-lived infected cells before viral integration ($\delta_1$), (ii) reduction of the death rate of productively infected cells ($\delta_2$), (iii) increasing the viral infection rate ($\beta$), or (iv) increasing the virus production rate ($p$). Administration of this type of depleting antibody has been shown to induce CD4+ T cell proliferation[20,27], thus we also allow for an effect of depletion increasing the CD4+ T cell proliferation rate ($r$). These effects were simulated by changing the corresponding parameters in Eq. (1) under depletion conditions to $(1-\xi_1)\delta_1$, $(1-\xi_2)\delta_2$, $(1+\xi_3)\beta$, $(1+\xi_4)p$ and $(1+\xi_5)r$ then estimating $\xi_i$, one at a time or in combination. We thus have 32 different models, from all $\xi_i = 0$ (our original model) to all $\xi_i \neq 0$ (indicating the CD8+ depletion influences all of the five parameters). Depending on the effect or combination of effects in each fit, we estimate the respective reduction or increase in each parameter ($\xi_i$). In addition, we estimated parameter distributions for $V_0, \delta_1, \delta_2, \omega, \beta, r, K$, the initial fraction of virus produced by short-lived infected cells $f_I$ and the pharmacological delay for raltegravir $\tau$ (see[44] for model details). We assumed the parameter $\omega$ follows a logit distribution to ensure values between 0 and 1. We fixed parameters $d = 0.01$ day⁻¹, $p = 4 \times 10^4$ virus/(cell day)[63], $k = 2.6$ day⁻¹, $k_m = 0.017$ day⁻¹, $\delta_m = 0.02$ day⁻¹ and $c = 23$ day⁻¹, as in Cardozo et al.[44] In these fits, $t = 0$ is the time of the first M-T807R1 administration. We assumed that the system in (1) is in steady state before $t = 0$, allowing us to obtain the values of $I_1(0) = \frac{\delta_2 c f_I V(0)}{kp}$, $M_1(0) = \frac{\delta_2 c(1-f_I)V(0)}{k_m p}$, $I_2(0) = \frac{cV(0)}{p}$,

$$T_I(0) = \frac{I_1(0)}{\frac{I_1(0)}{K} - \beta V(0)}(r(1 - \frac{I_1(0)}{kpK}) - k - \delta_1), \quad \beta_m T_M(0) = \frac{\delta_m + k_m}{V(0)}M_1(0) \quad \text{and}$$

$$\lambda = (d + \beta V(0) - r(1 - \frac{I_1(0) + T_I(0)}{K}))T_I(0).$$

We fit each model to the entire data set 10 times using random initial guesses of the parameters to be estimated. Each of these times, we estimated the log-likelihood ($\log \mathcal{L}$). For the case with highest likelihood ($\max(\log \mathcal{L})$) from the 10 fits, we then computed the corrected Akaike Information Criteria (AICc) for multivariable problems as $AICc = -2\max(\log \mathcal{L}) + 2m + \frac{2m(m+3)}{n-m-3}$, where $m$ is the number of parameters estimated and $n$ de number of data points from all animals[64]. We used AICc to compare models with different CD8$^+$ cell effects. Note that when we fit all the data together, we have 622 data points and 18–23 parameters depending on the model (from a model with no effect to one with all effects of CD8$^+$ depletion).

We also analyzed the structural identifiability of our model analytically[65] and calculated profile likelihoods[66–68] for each of the fitted parameters using Monolix (see Supplementary Note 2).

### Statistical analyses

We compared levels of different variables using linear mixed-effects models for repeated data within macaque (package lmerTest v 3.1-3 in R v 4.2.0), with the log10 transformation to stabilize the variance, or Mann–Whitney test when comparing values from different animals across treatment groups. When comparing more than two groups a Bonferroni correction was applied. A two-sided p-value below 0.05 was considered significant. Box plots (in Figs. 2, 3 and 4) represent the 25th and 75th percentiles (bottom and top edge of the box), the median (line across the box), whiskers extending from the edge of the box to the smallest (bottom) or largest (top) value no further than 1.5 times the interquartile range from the box's edges, and extreme values beyond that shown as crosses.

### Reporting summary

Further information on research design is available in the Nature Portfolio Reporting Summary linked to this article.

## Data availability

The raw data for all graphs generated in this study are provided in the Supplementary Information and Source Data file. Source data are provided with this paper.

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

## Acknowledgements

We gratefully acknowledge the veterinary support by G. Haret-Richter and T. Dunsmore. Raltegravir was generously provided by Merck Sharp & Dohme. The anti-CD8 alpha [MT807R1] antibody was provided by the NIH Nonhuman Primate Reagent Resource (P40 OD028116). We would also like to thank Mario Castro for valuable discussions regarding model structural identifiability. This work was supported by the National Institutes of Health (including the National Center for Research Resources, the National Institute for Allergy and Infectious Diseases, and the National Heart, Lung and Blood Institute) grants P01 AI169615 (A.S.P.), R01 AI152703 (R.M.R.), R01 AI104373 (R.M.R.), UM1 AI164561 (R.M.R.), R01

AI028433 (A.S.P.), R01 OD011095 (A.S.P.), R01 AI150500 (E.F.C.), R01 HL117715 (I.P.), R01 AI119346 (C.A.), R01 DK119936 (C.A.) and DK131476 (C.A.). B.B.P. and A.J.K. were supported in part by the NIH Training Grant T32 AI065380. A.J.K. was also supported in part by NIH Training Grant T32 AI060525. R.M.R. was also partly funded by grant PTDC/MAT-APL/31602/2017 from the Fundação para a Ciência e Tecnologia (Portugal). The funders had no role in study design, data collection and analysis, decision to publish, or preparation of the manuscript.

## Author contributions

Conception and study design: A.S.P., C.A., I.P., and R.M.R. Experimental work and data acquisition: B.B.P., C.X., D.M., T.H., K.R., R.S., A.K., C.A., and I.P. Data analyses: E.F.C., A.S.P., and R.M.R. Writing and revising: B.B.P., E.F.C., A.S.P., C.A., I.P., and R.M.R. Approval submitted version: B.B.P., E.F.C., C.X., D.M., T.H., K.R., R.S., A.K., A.S.P., C.A., I.P., and R.M.R.

## Competing interests

The authors declare no competing interests.
