## [Peer Review File · Nature Communications]

REVIEWER COMMENTS

Reviewer #1 (expertise in SIV infection, HIV natural control, virus replication):

Remarks to the Author:

Policicchio and colleagues used SIV-infected rhesus macaques receiving CD8-depleting antibody; the integrase inhibitor Raltegravir (RAL) + CD8-depleting antibody; or only RAL to help elucidating whether CD8+ T cells act through cytolytic and/or non-cytolytic mechanisms on SIV infection. The authors used viral dynamic and mathematical model. In summary, they found that CD8 T cells might exert both a cytolytic effect against infected cells prior to viral integration, and a non-cytolytic effect reducing viral production by infected cells after integration.

Recent batch of evidences suggest that CD8 T cells have relevant non-cytolytic activity in the context of HIV/SIV pathogenesis. Definitely, elucidating this question is of major relevance in the context of HIV cure/remission strategies. This paper is a relevant contribution to the field. However, the present study presents some limitations that must be addressed.

Major comments:

- 1. Prior studies have evoked that CD8 T cells might have cytolytic effects prior to productive virus replication. Here the authors assessed this question by using the integrase inhibitor RAL and CD8 T cell depletion. In addition, they adapted their previously proposed slow and rapid virus integration model to fit the data obtained. These fundamental differences from former studies must be clearly discussed to evidence the originality of the present work.**
- 2. One of the main caveats of CD8 depletion studies is the substantial increase in the levels of proliferation and activation of CD4 T cells, main target of the virus. Here the authors fitted the data on CD4 T cell proliferation to try to overcome this limitation. Yet, their results showed that it was a robust finding needed to explain the observed dynamics. CD4 T cell proliferation and activation peak at acute SIV-infection. Do the authors believe that the conclusions would be different if CD8 depletion was performed earlier after SIV-infection?**
- 3. The authors call "pre-integration" the data obtained following RAL therapy. This is true for new rounds of virus infection. However, at day 56 post-SIV infection, viral reservoir was already established and the pool of infected cells is present in blood and tissues. How to exclude (or include in the model) the fact that CD8 T cells could be killing productive CD4 T cells infected prior to RAL treatment?**
- 4. Considering the timing of CD8 depletion (day 56 post-infection), we can consider that the SIV-specific CD8 response is already contracting, therefore the pool of effector cells is decreasing, giving place to less differentiated memory CD8 T cells. Do the authors believe that differences in the distribution of SIV-specific CD8 T cell subpopulations with distinct functional profiles could impact the conclusions regarding cytolytic versus non-cytolytic activities?**
- 5. Data on natural HIV and SIV controllers have shown that the quality of CD8 T cells and the virus-specific CD8 T cell responses relies in the superior functional capacity of these cells to eliminate infected CD4 cells. In a model of SIV natural control, Madelain and colleagues (available in biorxiv) have shown that viral kinetics was best fitted using a model where the cytotoxic immune response progressively mounted up and reduced actively infected cells half-life. To the best of my knowledge, this is the only assessment of the impact of CD8 T cells on the longevity of productively SIV-infected cells in vivo in a model of natural control, where the role of CD8 T cells in controlling virus is determinant. Based on these evidences, the authors should discuss this point and clarify that the cytolytic effect observed prior to viral integration and the non-cytolytic effect**

reducing viral production by infected cells after integration are likely a hallmark of CD8 T cells from non-controllers individuals.

Minor comments:

- 1. Less pronounced CD8 depletion in the lymph nodes of RAL+CD8 depletion group. Is the difference from the CD8 depletion group significant? Please, comment potential implications.**
- 2. Frequency of NK cells seems very heterogeneous prior to CD8 depletion. Also increase in the % of NK cells were more frequently observed in the group receiving RAL. Do the authors think this might have an impact?**
- 3. Do the authors have experimental measurements of the cytolytic CD8 T cell activity and/or non-cytolytic soluble factors to fit the model and to provide additional support to the conclusions?**

Reviewer #2 (expertise in AIDS pathogenesis, prevention, and therapy; SIV and HIV infection):

This article by Policicchio et al., reports the results of an interesting NHP study aimed at addressing the role of cytolytic and non-cytolytic CD8+ T cell-mediated immune responses on HIV/SIV production, and in particular the possibility that a key effect of CD8+ T cells is to reduce virus production in a non-cytolytic fashion but through transcriptional silencing. The study uses an elegant in vivo model in which 20 rhesus macaques are infected with SIVmac251 and divided in three groups (CD8-depletion alone, integrase inhibitor and CD8-depleting antibody, and integrase inhibitor alone) and a mathematical model that includes infected cells pre- and post- viral DNA integration is applied to compare different immune mechanisms of virus suppression. The main finding of this study is that a model that includes both cytolytic CD8+ cell effects occurring before integration and non-cytolytic effects that reduce viral production best explain the viral profiles across all animals and groups.

Overall, this is very well conducted and truly important in vivo study that, while confirming the importance of CTL as a key mechanism underlying the antiviral effect of CD8+ T cells (even prior to virus integration), is consistent with and supporting of a large body of recent data emphasizing the role of CD8+ T cells in suppressing virus production through non cytolytic mechanisms such as transcriptional silencing (Cartwright et al., Immunity 2016; McBrien et al. Nature 2020; Zanoni et al., PLoS Pathogens 2020; Wallace J Immunol 2020; Mc Brien et al., J Virol 2021; Barbian et al., PLoS Pathogens 2022; etc). In this regard this study is very important in that provides theoretical and quantitative underpinnings from a well-designed in vivo study to change the paradigm on how CD8+ T cells exert an anti-HIV/Siv effect by suppressing virus production. I only have one major comment and a few minor clarifications, none of which require further experiments:

Major comment:

- 1. When using the word "non-cytolytic" to define the abrogation of virus suppression the authors are of course correct, but that term includes both non-cytolytic effect caused by secretion of soluble factors that could act as entry inhibitors (i.e., reduced virus infectivity) as well as non-cytolytic effect caused by suppression of virus production (i.e., transcriptional silencing). Since the latter mechanism but not the first fits with the reported data and the used mathematical model, the Authors should be more explicit in this regard, and (i) refer directly to "suppression of virus production" or "transcriptional silencing", and (ii) avoid the term "non-cytolytic" used tout court, such in the title, as it can be confusing and misleading (an alternative title could be, for instance: "CD8+ T-**

cells control infection by exerting both cytolytic effects and non-cytolytic suppression of virus production”, and the final sentence in the abstract could be “Our results suggest that CD8+ T-cells have both a cytolytic effect on infected cells before viral integration, and a direct, non-cytolytic effect on suppressing viral production through transcriptional and/or translational silencing”).

2. The Authors should explain a bit better why only 4 CD8 depletion alone animals were used.

3. Can the Authors confirm that the animals were never used before for studies of HIV/SIV vaccines?

4. Figure 2 panels A-B-C, I would “break” the line of the CD8 data prior to CD8 depletion, when applicable (A-B) as the current graphic representation gives the (wrong) impression that CD8 counts started falling before CD8 depletion, which is obviously not the case.

5. I would put figure 3 in the supplemental material and jump straight from the CD8 data to viral loads... but this may just be a matter of taste.

Reviewer #3 (expertise in mathematical modelling, HIV, virology):

This paper investigated the antiviral effect of CD8 T cells against SIV, in particular, “hypothetical” mechanism(s) of action, combining with SIV infection animal experiments and mathematical modeling. The authors evaluated three different regimens to SIV-infected rhesus macaques, that is, (1) CD8 depletion, (2) CD8 depletion + RAL and (3) RAL, and mainly measured viral load, CD8 T cells, NK/NKT cells and CD4 T cells in plasma and tissues. Then, employing 32 different mathematical models and fitting these models to time-course data of plasma viral load and CD4 T cells, they extracted the most “likely” mathematical model and discussed the assumption(s) for the possible antiviral effect of CD8 T cells from the statistical point of view (i.e., lowest AICc). Their main conclusion is that CD8 T cells kill infected cells before SIV DNA integration, which had not been observed before. There are several points which the authors need to be addressed to consider the publication.

1. First of all, their conclusions mainly rely on the results of best-fitted mathematical model to only two kinds of data, that is, plasma viral load and CD4 T cells (and they do not explicitly include the time-series data of CD8 T cells which are main player of this topics into their mathematical model and fit it to the data). The AIC analysis is not the definitive proof of mechanism(s) of action of antiviral effect of CD8 T cells. They, at least, must show experimental evidence that supports CD8 T cells kill infected cells before SIV DNA integration (which is the first observation and most appealing point in this study).

2. The fitting to Ki67 CD4 T cells is basically very poor. In particular, the dynamics of Ki67 CD4 T cells shows large variations among rhesus macaques with CD8 depletion and RAL.

3. Why and how CD8 T cells have a cytolytic role prior to viral integration? Why CD8 T cells do not have the role after the integration (or it just cannot be seen due to cytopathic effect)? What is the mechanism of this? The authors need to explain this point showing experimental data.

4. The estimation of “fraction reduction of death rate of SIV-infected short-lived cells pre-SIV-DNA integration during CD8 cell depletion” is 1. Is this reasonable? That is, can M-T807R1 almost 100% deplete CD8 T cells?

5. What is exact effect of M-T807R1? If there are other major effects except the reduction of CD8+ T-cells, then it cannot be concluded that the reason for the increase in virus is due to CD8+ T-cells, although they also discussed about it in "Discussion" of manuscripts. How dose-dependently M-T807R1 deplete CD8 cells?

We thank the editors and reviewers for carefully considering our study. Here we present a point-by-point reply (in blue font) to the useful suggestions of the reviewers, including an indication of all changes made.

Reviewer #1

Remarks to the Author:

Policicchio and colleagues used SIV-infected rhesus macaques receiving CD8-depleting antibody; the integrase inhibitor Raltegravir (RAL) + CD8-depleting antibody; or only RAL to help elucidating whether CD8+ T cells act through cytolytic and/or non-cytolytic mechanisms on SIV infection. The authors used viral dynamic and mathematical model. In summary, they found that CD8 T cells might exert both a cytolytic effect against infected cells prior to viral integration, and a non-cytolytic effect reducing viral production by infected cells after integration.

Recent batch of evidences suggest that CD8 T cells have relevant non-cytolytic activity in the context of HIV/SIV pathogenesis. Definitely, elucidating this question is of major relevance in the context of HIV cure/remission strategies. This paper is a relevant contribution to the field. However, the present study presents some limitations that must be addressed.

We thank the reviewer for taking the time to read our study carefully and provide comments, including the positive feedback that this is “a relevant contribution to the field”.

Major comments:

1. Prior studies have evoked that CD8 T cells might have cytolytic effects prior to productive virus replication. Here the authors assessed this question by using the integrase inhibitor RAL and CD8 T cell depletion. In addition, they adapted their previously proposed slow and rapid virus integration model to fit the data obtained. These fundamental differences from former studies must be clearly discussed to evidence the originality of the present work.

This is an important point, which we now explicitly indicate in the introduction and which indeed bolster the originality of our study. Specifically, in the last paragraph of the introduction, we state that the hypothesis had never been tested experimentally as we do in our study, and that the models are novel to account for CD8 depletion. These same points are reiterated in the discussion.

2. One of the main caveats of CD8 depletion studies is the substantial increase in the levels of proliferation and activation of CD4 T cells, main target of the virus. Here the authors fitted the data on CD4 T cell proliferation to try to overcome this limitation. Yet, their results showed that it was a robust finding needed to explain the observed dynamics. CD4 T cell proliferation and

activation peak at acute SIV-infection. Do the authors believe that the conclusions would be different if CD8 depletion was performed earlier after SIV-infection?

We agree with the reviewer that the issue of increased proliferation and activation of CD4 T cells is very important, although we note that the vast majority of CD8 depletion studies in the literature do not give this issue the necessary emphasis. Modeling of this effect is, we believe, another original contribution of our study.

As the reviewer states, all the best models that we tested indicated that accounting for this increase was statistically supported by the data. On the other hand, our main results were robust to this issue, in the sense that fitting models only to the viral load (without considering the CD4+ T cell proliferation) – see the supplementary model – provided the same overall conclusions for the effect of CD8 cells (apart from the effect of increased proliferation not in that model). It is possible that the results would be different, if the experiment was done earlier after infection. But the impact would depend on the differential increase in proliferation caused by CD8 depletion vs. the naturally elevated proliferation at acute infection, and we do not have good data on this. In addition, our approach could not be directly applied to acute infection, because we assume that the virus is approximately at steady state when we start treatment (as seen in Fig S1), which is not a good assumption for primary infection. For these reasons, we decided not to speculate on this issue in the current paper.

3. The authors call “pre-integration” the data obtained following RAL therapy. This is true for new rounds of virus infection. However, at day 56 post-SIV infection, viral reservoir was already established and the pool of infected cells is present in blood and tissues. How to exclude (or include in the model) the fact that CD8 T cells could be killing productive CD4 T cells infected prior to RAL treatment?

We are glad that the reviewer highlighted this point, because we may not have explained the model properly. In the model, we have states of pre- and post-integration that exist independently of RAL treatment or not. For example, the data from the “CD8 depletion group” (with CD8 depletion but no RAL) was fitted with this same model. When treatment with RAL is started, cells can be blocked at the pre-integration state, but other cells will have already integrated DNA and other will still proceed with integration (as RAL therapy is not 100% efficient – in the model and in reality). In addition, all the models we tested include the potential for CD8-induced killing of productively infected cells, i.e. after integration (whether RAL is present or not). We are following the virus (which comes only from infected cells with integration), and we implicitly include all productively infected cells, independently of when they were infected (before or after RAL). Indeed, we have used this same model to analyze data of treatment with reverse transcriptase inhibitors.

We have explained this point better in the Methods, when we describe the model, saying:

“We note that in these models, we separate the viral lifecycle in pre- and post-integration stages, because integrase inhibitor therapy allows us to probe these stages, however they exist independently of treatment or the type of treatment. Indeed, before we used the same model to analyze data from people treated only with reverse transcriptase inhibitors.”

4. Considering the timing of CD8 depletion (day 56 post-infection), we can consider that the SIV-specific CD8 response is already contracting, therefore the pool of effector cells is decreasing, giving place to less differentiated memory CD8 T cells. Do the authors believe that differences in the distribution of SIV-specific CD8 T cell subpopulations with distinct functional profiles could impact the conclusions regarding cytolytic versus non-cytolytic activities?

Again, this is a very interesting question, but we don't have any appropriate data that we could use to discuss this issue in more detail in the paper. It is possible that the nature of the CD8 effector function changes with time since infection, although we conducted our protocol as early as possible post-acute infection after the virus reached a quasi-steady state. We expect this to be a time when the CD8 response is still vigorous because of the high viral load. We note also that our results would apply to the majority of infected people, who start treatment after acute infection.

We now added in the discussion of the manuscript a sentence about the possible effect of the timing of the study, including citation to a relevant reference, which mentions the possibility of reduced cytolytic capacity of CD8+ cells after primary infection.

5. Data on natural HIV and SIV controllers have shown that the quality of CD8 T cells and the virus-specific CD8 T cell responses relies in the superior functional capacity of these cells to eliminate infected CD4 cells. In a model of SIV natural control, Madelain and colleagues (available in biorxiv) have shown that viral kinetics was best fitted using a model where the cytotoxic immune response progressively mounted up and reduced actively infected cells half-life. To the best of my knowledge, this is the only assessment of the impact of CD8 T cells on the longevity of productively SIV-infected cells in vivo in a model of natural control, where the role of CD8 T cells in controlling virus is determinant. Based on these evidences, the authors should discuss this point and clarify that the cytolytic effect observed prior to viral integration and the non-cytolytic effect reducing viral production by infected cells after integration are likely a hallmark of CD8 T cells from non-controllers individuals.

We thank the reviewer for referring us to this paper. In fact, natural control of infection, especially in natural hosts of SIV is a particular interest of ours, and we published on the topic of infected cell half-lives in Africa-green monkey SIV infection (a natural host model) and showed that indeed they are very

short-lived (Pandrea et al. J Virol 82: 3713 (2008)). We also note that our current study does demonstrate an important effect of CD8+ cells, by the two mechanisms cytolytic and non-cytolytic, acting at different times. The paper of Madelain et al is very interesting, however they only test models that included a cytolytic effect, and do not make any comparisons with putative non-cytolytic mechanisms. Still, it is possible that the balance between cytolytic and non-cytolytic mechanisms is different in controllers vs. non-controllers (as also discussed in the study by Chowdhury et al J Virol 89: 8677 (2015)), and we now have added a comment on this in the discussion and cite these two papers.

“Some authors have argued that in models of natural control of SIV infection, CD8 cytolytic mechanisms are preponderant. Although those authors have not analyzed their data in as much detail as we do here, we can’t exclude the possibility that our results apply mostly to non-controlled SIV infection, which is the relevant model to the majority of HIV cases.”

Minor comments:

1. Less pronounced CD8 depletion in the lymph nodes of RAL+CD8 depletion group. Is the difference from the CD8 depletion group significant? Please, comment potential implications.

Yes, this difference is statistically significant, and we now indicate this in the section where we discuss potential limitations in the Discussion. It is possible that this lower depletion of CD8 in the RAL (and depleted) group is due to the reduction in viral load with treatment vs. the untreated CD8 depletion-only group. We think that, if anything, the less than perfect depletion in the lymph nodes would make it more difficult to see a difference between the RAL treated with and without CD8 depletion. The fact that we do see a difference in viral kinetics and this signal is enough to statistically differentiate our models indicates that the results are robust.

2. Frequency of NK cells seems very heterogeneous prior to CD8 depletion. Also increase in the % of NK cells were more frequently observed in the group receiving RAL. Do the authors think this might have an impact?

Because NK cells express CD8alpha, they are also affected by the depleting antibody, and this helps explain the higher frequencies in the RAL-only group. This is a potential limitation of our study, and that is why we included both the supplementary figure and a discussion of this possible issue in the Discussion. We note that there are many, many papers in the literature using the same protocol as ours to analyze the effect of CD8+ T cells using the same antibody, in spite of this potential caveat.

3. Do the authors have experimental measurements of the cytolytic CD8 T cell activity and/or non-cytolytic soluble factors to fit the model and to provide additional support to the conclusions?

Unfortunately, we don't have this type of data.

Reviewer #2:

This article by Policicchio et al., reports the results of an interesting NHP study aimed at addressing the role of cytolytic and noncytolytic CD8+ T cell-mediated immune responses on HIV/SIV production, and in particular the possibility that a key effect of CD8+ T cells is to reduce virus production in a non-cytolytic fashion but through transcriptional silencing. The study uses an elegant in vivo model in which 20 rhesus macaques are infected with SIVmac251 and divided in three groups (CD8-depletion alone, integrase inhibitor and CD8-depleting antibody, and integrase inhibitor alone) and a mathematical model that includes infected cells pre- and post- viral DNA integration is applied to compare different immune mechanisms of virus suppression. The main finding of this study is that a model that includes both cytolytic CD8+ cell effects occurring before integration and noncytolytic effects that reduce viral production best explain the viral profiles across all animals and groups.

Overall, this is very well conducted and truly important in vivo study that, while confirming the importance of CTL as a key mechanism underlying the antiviral effect of CD8+ T cells (even prior to virus integration), is consistent with and supporting of a large body of recent data emphasizing the role of CD8+ T cells in suppressing virus production through non cytolytic mechanisms such as transcriptional silencing (Cartwright et al., Immunity 2016; McBrien et al. Nature 2020; Zanoni et al., PLoS Pathogens 2020; Wallace J Immunol 2020; Mc Brien et al., J Virol 2021; Barbian et al., PLoS Pathogens 2022; etc). In this regard this study is very important in that provides theoretical and quantitative underpinnings from a well-designed in vivo study to change the paradigm on how CD8+ T cells exert an anti-HIV/Siv effect by suppressing virus production. I only have one major comment and a few minor clarifications, none of which require further experiments:

We thank the reviewer for the careful assessment of our study and for the extremely positive evaluation of our work.

Major comment:

1. When using the word "non-cytolytic" to define the abrogation of virus suppression the authors are of course correct, but that term includes both non-cytolytic effect caused by secretion of soluble factors that could act as entry inhibitors (i.e., reduced virus infectivity) as well as non-cytolytic effect caused by suppression of virus production (i.e., transcriptional silencing). Since the latter mechanism but not the first fits with the reported data and the used mathematical model, the Authors should be more explicit in this regard, and (i) refer directly to "suppression

of virus production” or “transcriptional silencing”, and (ii) avoid the term “noncytolytic” used tout court, such in the title, as it can be confusing and misleading (an alternative title could be, for instance: “CD8+ T-cells control infection by exerting both cytolytic effects and non-cytolytic suppression of virus production”, and the final sentence in the abstract could be “Our results suggest that CD8+ T-cells have both a cytolytic effect on infected cells before viral integration, and a direct, non-cytolytic effect on suppressing viral production through transcriptional and/or translational silencing”).

We agree with the reviewer that this clarification is important. We have made the suggested changes to the title, running head, abstract, and we changed the manuscript in many other places, especially the Discussion, to emphasize “suppression of virus production” rather than the more general “non-cytolytic”.

The new title is “CD8+ T-cells control infection by both cytolytic effects and non-cytolytic suppression of virus production at different stages of SIV infection”.

2. The Authors should explain a bit better why only 4 CD8 depletion alone animals were used.

We include a better justification in the Methods section. Essentially the simple depletion experiment has been done many times in the context of SIV infection, and for animal welfare issues (and cost) we just needed to make sure that our depletion was equivalent to historical controls and so that the protocol was working properly.

3. Can the Authors confirm that the animals were never used before for studies of HIV/SIV vaccines?

Yes, these animals were never used before for any other scientific studies. We added a sentence to this effect in the Methods, where we describe the animals used.

4. Figure 2 panels A-B-C, I would “break” the line of the CD8 data prior to CD8 depletion, when applicable (A-B) as the current graphic representation gives the (wrong) impression that CD8 counts started falling before CD8 depletion, which is obviously not the case.

We thank the reviewer for this suggestion, and we implemented in the revised figure 2.

5. I would put figure 3 in the supplemental material and jump straight from the CD8 data to viral loads... but this may just be a matter of taste.

Indeed, this is what we had done in a previous version of the manuscript. But, because we use the CD4+Ki67+ data in our model fits, which was not done in previous studies, and proliferation was an important mechanism to consider, we prefer to leave the presentation as it is now.

Reviewer #3:

This paper investigated the antiviral effect of CD8 T cells against SIV, in particular, “hypothetical” mechanism(s) of action, combining with SIV infection animal experiments and mathematical modeling. The authors evaluated three different regimens to SIV-infected rhesus macaques, that is, (1) CD8 depletion, (2) CD8 depletion + RAL and (3) RAL, and mainly measured viral load, CD8 T cells, NK/NKT cells and CD4 T cells in plasma and tissues. Then, employing 32 different mathematical models and fitting these models to time-course data of plasma viral load and CD4 T cells, they extracted the most “likely” mathematical model and discussed the assumption(s) for the possible antiviral effect of CD8 T cells from the statistical point of view (i.e., lowest AICc). Their main conclusion is that CD8 T cells kill infected cells before SIV DNA integration, which had not been observed before. There are several points which the authors need to be addressed to consider the publication.

We thank the reviewer for critical reading of the manuscript and noting its originality.

1. First of all, their conclusions mainly rely on the results of best-fitted mathematical model to only two kinds of data, that is, plasma viral load and CD4 T cells (and they do not explicitly include the time-series data of CD8 T cells which are main player of this topics into their mathematical model and fit it to the data). The AIC analysis is not the definitive proof of mechanism(s) of action of antiviral effect of CD8 T cells. They, at least, must show experimental evidence that supports CD8 T cells kill infected cells before SIV DNA integration (which is the first observation and most appealing point in this study).

We agree with the reviewer that statistical support is not proof of mechanism, although we believe that statistical comparisons of the multiple mechanisms is a very important result. We note, for example, that previous studies analyzing viral decay with reverse transcriptase inhibitors (only) with or without CD8 depletion did not find any statistical differences in the decay rates (Klatt et al. PLoS Pathogens 6: e1000747 (2010) and Wong et al PLoS Pathogens 6: e1000748 (2010)).

We also agree with the reviewer that experimental evidence supporting “CD8 T cells killing infected cells before SIV DNA integration” is very important, and we should have included an expanded discussion of

this in our paper. Although we didn't do these experiments ourselves, there are several important papers in the literature demonstrating that this is possible.

Sacha et al. ("Gag-specific CD8+ T lymphocytes recognize infected cells before AIDS-virus integration and viral protein expression", *J Immunol* 178: 2746 (2007)) did the exact experiment that the reviewer mentions. They used cells from Indian origin rhesus macaques (as in our study) to show that CD8+ T cells recognize Gag-derived epitopes from SIV infection very early after infection of a cell, they say "Two different Gag-specific CD8+ T cell clones restricted by two different MHC-I molecules recognized their cognate epitopes as early as 2 h postinfection with recognition peaking at ~6h", i.e., before viral integration. Moreover, they went further and showed that these epitopes are derived from the incoming virus, by using non-infectious virus or preventing reverse transcription and showing that these systems still elicited similar Gag-specific CD8+ T cells with similar kinetics over the first 6h. Finally, they directly showed that "Gag-specific CD8+ T cells eliminate infected CD4+ T cells early after infection", and state that "The Gag-specific clone, Gag71–79GY9, eliminated many of the infected cells by 6 h postinfection, reducing the frequency of Gag p27+CD4+ T cells at 6 h postinfection from 66% (no CD8+ T cell control) to 34.6%". In a second similar paper, Sacha et al ("Pol-Specific CD8+ T Cells Recognize Simian Immunodeficiency Virus-Infected Cells Prior to Nef-Mediated Major Histocompatibility Complex Class I Downregulation" *J Virol* 81: 11703 (2007)) showed essentially the same early killing for Pol-derived epitopes. Note that in these studies, for other epitopes derived from Tat, Nef or Env, which are mostly generated in productively infected cells (after integration), CD8+ T cell effects first occurred only after 12-18h postinfection.

In another later paper by a different group, Kloverpris et al ("Early Antigen Presentation of Protective HIV-1 KF11Gag and KK10Gag Epitopes from Incoming Viral Particles Facilitates Rapid Recognition of Infected Cells by Specific CD8+ T Cells", *J Virol* 87: 2628 (2013)) show essentially the same results of early recognition, activation and killing by CD8+ specific T cells recognizing early presented epitopes in HIV infected human CD4+ T cells (and other targets) before viral integration and before protein synthesis. In this paper, again, they use a variety of controls and timing experiments to show these results.

Finally, an older paper using cell lines infected with HIV-1, Yang et al ("Efficient Lysis of Human Immunodeficiency Virus Type 1-Infected Cells by Cytotoxic T Lymphocytes", *J. Virol* 70: 5799 (1996)), while not looking directly at the issue of timing of CTL killing, included experiments that led them to state "These data suggest that recognition of HIV-1-infected cells by CTL occurs early in the course of infection, prior to the production of significant amounts of virions".

These beautiful experiments are full studies of their own right and confirm that "CD8 T cells kill infected cells before SIV DNA integration" as requested by the reviewer. We did not repeat them, since those results are presented in exquisite detail and (at least Sacha et al) use an in vitro system (macaque cells and virus) that is very similar to our in vivo experimental study.

We have now added detailed description and contextualization (4th paragraph) of these studies in the "Discussion" of our paper.

2. The fitting to Ki67 CD4 T cells is basically very poor. In particular, the dynamics of Ki67 CD4 T cells shows large variations among rhesus macaques with CD8 depletion and RAL.

We recognize that there is quite a bit of variability in the dynamics of Ki67+ CD4+ T cells among the macaques, as is very often the case for the dynamics of immune populations (by contrast with virus dynamics). This makes the fits look poor. The variability probably is due to stochastic variation between the macaques over time, and a reason why previous studies tended to ignore the effect of CD8 depletion on CD4+ T cell proliferation. For this reason, we also presented fits of a simpler model not considering the dynamics of Ki67+CD4+ T cells. These results correspond to a sensitivity analyses (to the model and data used) to our primary results and show full consistency in terms of the mechanisms of action of CD8+ cells.

We now have added more detail on this point in the Discussion, and present in the supplementary material more details of the alternate fits without considering the Ki67+ CD4+ T cell data, including new figures S5 and S6, with the individual fits and the results of model selection. In the new text of the Discussion, we say:

“Indeed, our results showed that an increase in CD4+ T-cell proliferation rate after CD8+ cell depletion was a robust finding, in spite of the variability seen in the dynamics of Ki67 CD4+ T cells (Figure S4). As a sensitivity analyses to our model formulation and to the use of this variable data, we also fitted a model only to the viral load without considering the Ki67 CD4+ T-cell data (Figure S5), and reached the same conclusions on the effect of CD8+ T cells, namely that they exert a cytolytic effect before viral integration and non-cytolytic suppression of viral production.”

3. Why and how CD8 T cells have a cytolytic role prior to viral integration? Why CD8 T cells do not have the role after the integration (or it just cannot be seen due to cytopathic effect)? What is the mechanism of this? The authors need to explain this point showing experimental data.

This is the question that we sought to analyze with our study. Indeed, we allow CD8+ cells to have cytolytic effects before and after integration. What we show is that the best description of the data does not include a cytolytic effect after the cells become productively infected. Most likely the reviewer is correct, and this is because at that stage viral cytopathic effects are more important and obscure any effect of CD8+ induced killing. It is possible that CD8+ cells still have a cytolytic effect, but it is minor compared to the viral-induced death.

That the lifespan of productively infected cells is not affected by CD8+ T-cells, thus indicating no (or a minor) role for cytolytic effect of these cells, is supported by multiple lines of experimental evidence. Two previous studies (Klatt et al. PLoS Pathogens 6: e1000747 (2010) and Wong et al PLoS Pathogens 6: e1000748 (2010)) with a CD8 depletion experiment in SIV reached the same conclusion, i.e., no cytolytic effect after infected cells become productive. Other studies with different approaches in the setting of HIV treatment also showed that i) there was no difference in the death rates of productively infected cells early vs. late post HIV infection, with the idea that cytolytic activity would be better preserved early vs. late (Phillips et al. “In vivo HIV-1 replicative capacity in early and advanced infection” AIDS 13: 67 (1999)) and the more recent Kilby et al. “Treatment response in acute/early infection versus advanced AIDS: equivalent first and second phases of HIV RNA decline”, AIDS 22: 957 (2008)); or ii) for people with protective vs. non-protective HLA alleles, with the idea that cytolytic activity would be superior with

protective alleles (Spits et al “The presence of protective cytotoxic T lymphocytes does not correlate with shorter lifespans of productively infected cells in HIV-1 infection” AIDS 30: 9 (2016)).

We now, in addition to mentioning those studies in the Introduction, clarify this question in the Discussion (1st paragraph) mentioning explicitly the hypothesis of direct viral cytopathic effect obscuring any putative cytolytic effect of CD8+ cells.

4. The estimation of “fraction reduction of death rate of SIV-infected short-lived cells pre-SIV-DNA integration during CD8 cell depletion” is 1. Is this reasonable? That is, can M-T807R1 almost 100% deplete CD8 T cells?

M-T807R1 is very good at depleting CD8+ T cells (as shown in our paper and previous publications). However, there can be other explanations for the estimate of 1, even in the absence of 100% depletion. One possibility is that a critical number of CD8+ cells are needed to exert significant control, and when these cells fall below that threshold control is almost completely lost. We note that in the simpler model, without accounting for proliferation of target cells, the estimate for the fraction reduction of the death rate of infected cells pre-SIV DNA integration was still large at 65%-70%, but not quite 100%. We now present this other model in more detail, with a separate paragraph in the results, and new figures S5 and S6, and also mention the results of this model in the Discussion.

5. What is exact effect of M-T807R1? If there are other major effects except the reduction of CD8+ T-cells, then it cannot be concluded that the reason for the increase in virus is due to CD8+ T-cells, although they also discussed about it in “Discussion” of manuscripts. How dose-dependently M-T807R1 deplete CD8 cells?

M-T807R1 is a standard monoclonal antibody reagent provided by the National Institutes of Health through the Nonhuman Primate Reagent Resource for depletion of CD8+ cells (<https://www.nhpreagents.org/Store/ProductID/44>). It is a rhesusized version of a previously developed murine-human chimeric CD8 depleting antibody, which is described in detail in Schmitz et al. American J Pathology 154: 1923 (1999), including issues of mechanism of action. We did not study dose-dependency of this antibody, but used the amount of antibody (50mg/kg) that has been shown in the literature to achieve near complete depletion of CD8+ cells (as we also show in our paper). In addition to depleting CD8+ T cells, this intervention also increases the proliferation of CD4+ T cells, which can be due to the large depletion of CD8 or a putative direct effect of the Ab. This is why we take this effect into account in our model. As mentioned in the Discussion, M-T807R1 depletes CD8+ cells, which in addition to T cells may include NK cells, as we show in Figure S3, and mention in the Discussion as a possible limitation. However, the accepted idea of the scientific community is that the main effect in infection is

through depletion of CD8+ T cells, as attested by many papers using a protocol similar to ours to assess CD8 control of many different infections. As examples, we can mention i) SIV in natural hosts (Gaufin et al. "Experimental depletion of CD8+ cells in acutely SIVagm-infected African Green Monkeys results in increased viral replication" *Retrovirology* 7: 42 (2010)), ii) Ebola (Sullivan et al. "CD8+ cellular immunity mediates rAd5 vaccine protection against Ebola virus infection of nonhuman primates" *Nat Med* 17: 1128 (2011)), iii) smallpox (Edghill-Smith et al "Smallpox vaccine-induced antibodies are necessary and sufficient for protection against monkeypox virus", *Nat Med* 11: 740 (2005)), iv) measles (Permar et al. "Limited contribution of humoral immunity to the clearance of measles viremia in rhesus monkeys" *J Infect Dis* 190: 998 (2004)), v) hepatitis C virus (Shoukry et al. "Memory CD8+ T cells are required for protection from persistent hepatitis C virus infection" *J Exp Med* 197: 1645 (2003)), vi) hepatitis B virus (Thimme et al "CD8(+) T cells mediate viral clearance and disease pathogenesis during acute hepatitis B virus infection" *J Virol* 77: 68 (2003)), vii) SARS-CoV-2 infection (McMahan et al. "Correlates of Protection Against SARS-CoV-2 in Rhesus Macaques" *Nature* 590: 630 (2021)), viii) influenza (Carroll et al. "Memory B Cells and CD8+ Lymphocytes Do Not Control Seasonal Influenza A Virus Replication after Homologous Re-Challenge of Rhesus Macaques" *PLoS One* 6: e21756 (2011)) and ix) many SIV studies (e.g., Schmitz et al. "Control of viremia in simian immunodeficiency virus infection by CD8+ lymphocytes" *Science* 283: 857–860 (1999); Bruel et al. "Long-term control of simian immunodeficiency virus (SIV) in cynomolgus macaques not associated with efficient SIV-specific CD8+ T-cell responses" *J Virol* 89: 3542-3556 (2015); Cartwright et al. "CD8(+) Lymphocytes Are Required for Maintaining Viral Suppression in SIV-Infected Macaques Treated with Short-Term Antiretroviral Therapy" *Immunity* 45: 656-658 (2016); McBrien et al. " Robust and persistent reactivation of SIV and HIV by N-803 and depletion of CD8+ cells" *Nature* 578: 154-159 (2020), among many others.

Most these studies use (essentially) the same protocol as we do, including the same dose of antibody, to specifically address the effect of CD8+ T cells in the control of all these infections, concluding that either these cells are critical or not in each case. In this sense, our experimental protocol is just following the best practices in the literature.

Reviewers' Comments:

Reviewer #1:

Remarks to the Author:

The manuscript is significantly improved after the first round of revisions. The authors have addressed all my comments and the requested information was included in the new version of the manuscript. I have no additional comments.

Reviewer #2:

Remarks to the Author:

The Authors have satisfactorily addressed my comments and I believe the study is now ready for publication. Congratulations to the Authors for this interesting, important, and very well-designed body of work.

Reviewer #3:

Remarks to the Author:

The authors have provided a reasonable response to the comments I made regarding their manuscript, particularly with regards to my previous comments 1, 2, and 5. However, I still have two major concerns with regards to my previous comments 3 and 4.

1. Regarding my previous comment (number 3), I would like to further discuss their conclusion that CD8 T cells have a cytolytic effect prior to viral integration, and a non-cytolytic effect in suppressing virus production after viral integration. I am curious about the mechanism behind this "switch" in the major role of CD8 T cells at different stages (not a statistical point of view). Why is it necessary for CD8 T cells to switch their major role in controlling SIV infection?". Please discuss this in their Discussion.

2. Regarding my previous comment (number 4), their reply is not perfect. My main concern regarding this study is the identifiability of parameter estimation. The data sets used in this study consisted of longitudinal viral load and CD4 T cell measurements obtained from a small number of macaques. The authors estimated numerous parameters in their mathematical model, while also considering variations among the macaques.

2-a. One point of concern is their estimation of the reduction of death rate of short-lived cells before viral integration during CD8 cell depletion, which is 100% ($\zeta_1=1$) even for models with higher ΔAICc (Supplementary Table 1). This suggests that the estimation of the reduction may have failed. In contrast, the estimations with their simpler model, which did not account for the proliferation of target cells, ranged from 0.64-0.98 (Supplementary Table 2). This indicates that the estimation of ζ_1 differs among models with higher ΔAICc . I am curious about the reason for this reduction compared to $\zeta_1=1$ under the assumption of non-proliferation.

2-b. Additionally, they reported ΔAICc instead of AICc in Supplementary Table 1 and Table 2, which makes it difficult to compare their original model and simpler model. It would be helpful if they also included AICc .

2-c. To address the issue of identifiability of parameter estimation, they could test the structural identifiability of all parameter estimates by calculating the profile likelihood [pmid:27588423, pmid:19505944] using the dMod package [<https://www.jstatsoft.org/article/view/v088i10>] in R Statistical Software."

Reply to reviewer #3 (the reviewer's comments are in black and our responses in blue).

The authors have provided a reasonable response to the comments I made regarding their manuscript, particularly with regards to my previous comments 1, 2, and 5. However, I still have two major concerns with regards to my previous comments 3 and 4.

We thank the reviewer for these comments, and we agree that the manuscript has improved by the careful reading of all reviewers. We have also carefully considered the new comments and provide more information and new figures to address further the reviewer's concerns.

1. Regarding my previous comment (number 3), I would like to further discuss their conclusion that CD8 T cells have a cytolytic effect prior to viral integration, and a noncytolytic effect in suppressing virus production after viral integration. I am curious about the mechanism behind this "switch" in the major role of CD8 T cells at different stages (not a statistical point of view). Why is it necessary for CD8 T cells to switch their major role in controlling SIV infection?". Please discuss this in their Discussion.

This is indeed not a statistical issue, but a biological one and we thank the reviewer for pointing this out. Essentially, before integration the cell is already expressing viral epitopes that mark the cell as target for the CD8+ T-cell mediated response. And so the CD8+ T cells can kill infected cells. However, before integration there is no copying of viral DNA or de novo expression of viral proteins, so there is no target for the CD8 effector mechanisms to prevent production. This latter effect can only occur once the virus is poised to make new copies of itself, which is possible only after integration. As mentioned, this reflects the biological lifecycle, not statistics. At this point, after integration, there are, at least, three competing processes: i) viral production and its inhibition by the CD8+ immune response; ii) cell death caused by virus cytolytic effects; and iii) cell death caused by the CD8+ immune response. What we found in this study, by applying our dynamic model is that the signal for inhibition of viral production is the major one for the effect of CD8+ T cells. This aspect is a statistical result. As the reviewer mentioned in the first round of comments, it is possible that this is related to faster killing by the virus itself. And as we mentioned also before, this is consistent with other multiple types of data.

Recently, a new experimental paper studying latency in HIV infection also found that CD8+ cells have an effect of driving down viral expression, consistent with our results, and we also now include this reference in our discussion.

We included additional information in the discussion (lines 306-311) regarding this along the lines of the explanation above.

2. Regarding my previous comment (number 4), their reply is not perfect. My main concern regarding this study is the identifiability of parameter estimation. The data sets used in this study consisted of longitudinal viral load and CD4 T cell measurements obtained from a small number of macaques. The authors estimated numerous parameters in their mathematical model, while also considering variations among the macaques.

Please see below for our answers to the specific concerns of the reviewer.

2-a. One point of concern is their estimation of the reduction of death rate of short-lived cells before viral integration during CD8 cell depletion, which is 100% ($\xi_1=1$) even for models with higher ΔAICc (Supplementary Table 1). This suggests that the estimation of the reduction may have failed. In contrast, the estimations with their simpler model, which did not account for the proliferation of target cells, ranged from 0.64-0.98 (Supplementary Table 2). This indicates that the estimation of ξ_1 differs among models with higher ΔAICc . I am curious about the reason for this reduction compared to $\xi_1=1$ under the assumption of non-proliferation.

We believe that what supplementary Table 1 (for the more complex model) shows is that whenever we allow an effect of CD8 depletion on the death rate of infected cells before integration there is consistency of estimates, and the model fitting makes $\xi_1=1$. We agree that 100% reduction is drastic, and possibly due to difficulty in estimating this parameter. We now include the profile likelihood curves as requested by the reviewer and we see that the best estimates of this parameter (last panel in new figure S7) are indeed very close to 1, but (as the reviewer suspected) this parameter is difficult to estimate.

On the other hand, in the simpler model, the estimate of ξ_1 in the different fits is not the same, but it is consistently high and more constrained, as the reviewer mentioned (note that there was a typo and in supplementary table where it said 0.64 it should be 0.74). This implies that in the simpler model this parameter is easier to estimate (see corresponding panel in new figure S8 with the profile likelihoods for this simpler model).

Together these results indicate that depletion of CD8+ cells reduces the death rate of infected cells before integration. This reduction is likely to a high level, more than 70% ($\xi_1>0.7$), but the exact value may be difficult to estimate.

We added more information about these points in the manuscript, including new sections in the supplementary materials, the new profile likelihood figures, referring to these in the main text and removing the statement in the main text that the reducing was “consistently close to 100%” (in lines 228-229).

2-b. Additionally, they reported ΔAICc instead of AICc in Supplementary Table 1 and Table 2, which makes it difficult to compare their original model and simpler model. It would be helpful if they also included AICc .

We have now, for completeness, included the absolute AICc for the best model as a footnote of the corresponding tables (supplementary tables 1 and 2), allowing the calculation of all absolute AICc .

However, we note that we can't compare the AICc of the original and simpler models, because they use different data sets (viral load and Ki67 vs. just viral load), and one can only compare AICc for different models on the same data set. This is the reason why we didn't compare the two models in terms of fits.

2-c. To address the issue of identifiability of parameter estimation, they could test the structural identifiability of all parameter estimates by calculating the profile likelihood [pmid:27588423, pmid:19505944] using the dMod package [<https://www.jstatsoft.org/article/view/v088i10>] in R Statistical Software."

We thank the reviewer for this suggestion. Indeed, we should have shown the profile likelihoods before. Monolix can perform this calculation, although it is quite slow, and we have now presented these figures, and added a discussion about them in the supplementary methods. In addition, we analyzed analytically the structural identifiability of our model using the algorithm provided in Castro et al. PLoS Comp Biol, which is similar to the ideas in the papers indicated by the reviewer – this forms a new section in the supplementary material. We note, however, that structural identifiability in standard fitting is different from mixed-effects model fitting, as we do in this study. This is because we are estimating the distribution of the parameters, and not just the average. This is discussed briefly towards the end of the discussion in Castro et al..

We note that we weren't able to use the package suggested by the reviewer because of compiling issues that we were not able to resolve. Nevertheless, we also believe that we need to use algorithms appropriate for mixed-effects models, which Monolix is ideally suited to fit.

As mentioned, we added the new figures and the new sections, together with appropriate references, both in the supplementary materials and the main text.

Reviewers' Comments:

Reviewer #3:

Remarks to the Author:

Now all of my comments have been properly addressed. Congratulations!